# Intestinal Microecological Mechanisms of Aflatoxin B1 Degradation by Black Soldier Fly Larvae (*Hermetia illucens*): A Review

**DOI:** 10.3390/ani15223351

**Published:** 2025-11-20

**Authors:** Qiwen Yuan, Jing Xia, Chaorong Ge, Huaiying Yao

**Affiliations:** 1Hubei Key Laboratory of Microbial Transformation and Regulation of Biogenic Elements in the Middle Reaches of the Yangtze River, School of Environmental Ecology and Biological Engineering, Wuhan Institute of Technology, Wuhan 430205, China; qiwenyuan24@126.com (Q.Y.); chaorongge@163.com (C.G.); 2State Key Laboratory of Green and Efficient Development of Phosphorus Resources, Wuhan Institute of Technology, 206 Guanggu 1st Road, Wuhan 430205, China

**Keywords:** aflatoxin B1, black soldier fly larvae, intestinal microorganisms, biodegradation, microecological mechanisms

## Abstract

The pervasive contamination of agricultural supply chains by aflatoxin B1 directly endangers human health and animal welfare, which has established it as a class 1 carcinogen in international public health initiatives. Black soldier fly (*Hermetia illucens*) larvae, as saprophytic insects, hold complex and functionally diverse microbial communities in their intestinal tracts that function as a microbial reactor, providing a novel solution for mitigating aflatoxin B1 contamination. Additionally, it can tolerate the presence of multiple mycotoxins commonly found in contaminated substrates, and they are capable of efficiently detoxifying aflatoxin B1 through the synergistic effect of its detoxifying enzyme system and intestinal microorganisms. Therefore, the application of the black soldier fly larval intestinal microbial consortium for aflatoxin B1 biodegradation demonstrates significant potential for industrial implementation owing to its characteristics of low cost, high efficiency, safety, and sustainability.

## 1. Introduction

Aflatoxin (AF) is a mycotoxin with strong carcinogenicity and acute toxicity and is produced primarily by *Aspergillus flavus* and *Aspergillus parasiticus* [1]. Currently, more than 20 structurally distinct aflatoxin derivatives have been identified, including aflatoxins B1, B2, G1, G2, M1, and M2 (AFB1, AFB2, AFG1, AFG2, AFM1, and AFM2, respectively) [1]. Among them, AFB1 is primarily found in food crops, including peanuts, corn, wheat, rice, nuts, and seeds, and is extremely carcinogenic, teratogenic, neurotoxic, and immunotoxic to animals and humans [1,2,3]. Owing to these toxic properties, AFB1 has been designated as a class 1 carcinogen by the World Health Organization and remains a top concern in food safety and public health [2]. For example, in tropical and subtropical countries such as Kenya, India, and Tanzania, high-temperature and high-humidity climates make crops vulnerable to AF contamination (primarily AFB1), and risk prevention and control systems are still inadequate, leading to multiple incidents of AF poisoning [4,5,6]. Notably, the moldy corn contamination in Kenya in 2004 caused 317 poisonings and 125 deaths, making it the most serious AF poisoning incident on record [7]. Therefore, the majority of nations and jurisdictions worldwide have instituted rigorous permissible threshold concentrations (20 μg/kg) for AFB1 in food and feed to prevent its spread in the food supply chain and the consequent threat to public health safety [8].

The fundamental architecture of AFB1 comprises a difuran ring and a coumarin skeleton (Figure 1), and the molecule contains three active sites: the double bond between the eighth and ninth carbon atoms in the furan ring (Site 1), the coumarin-derived lactone ring (Site 2), and the substituent on the cyclopentenone ring (Site 3) [9]. Site 1 is regarded as the key structural site for inducing toxicity. Under the catalysis of hepatic cytochrome P450 monooxygenase (CYP450), opening of the double bond in site 1 generates the highly reactive intermediate AFB1-8,9-epoxide (AFBO), which induces genotoxicity and carcinogenic effects [3]. The coumarin lactone ring serves as the active site where AFB1 undergoes hydrolytic reactions, while the different substituents on the cyclopentenone ring can modulate its toxicity [9]. Therefore, destroying the active sites of AFB1 to block its toxic effects may be the key to achieving effective detoxification. Moreover, the difuran ring and coumarin skeleton of AFB1 form a nearly planar and rigid structure with a conjugated π–electron system [10]. Via π–electron conjugation, the structure induces electron delocalization, significantly reducing the molecular internal energy and enhancing the overall thermodynamic stability [10]. Furthermore, the activation energy for the cleavage of the double bond in site 1 within the AFB1 molecule is high, and the energy provided by the common high-temperature heating methods in food processing (e.g., pasteurization) is insufficient to break this chemical bond; thus, AFB1 is not easily degradable [10,11]. In addition, AFB1 exhibits the characteristic of limited aqueous solubility, pronounced lipid solubility, and high stability in the presence of strong acids and strong bases (pH = 3–10), facilitating its persistence in food matrices and subsequent spread through the food chain [12]. After being absorbed by the human gastrointestinal tract, AFB1 exerts biological toxicity in tissues and organs via the blood circulation [12]. Consequently, it is critically important to develop green, secure, and efficient detoxification strategies for eliminating AFB1 residues from the environment.

The present review summarizes the studies on the detoxification methods for AFB1 degradation, with a particular focus on microbial degradation processes, and the research progress regarding the sequencing of core intestinal microorganisms in BSFL, reported in the databases, such as National Center for Biotechnology Information (NCBI, https://www.ncbi.nlm.nih.gov/gdv/ (accessed on 12 November 2025)), PubMed (MEDLINE) database (https://pubmed.ncbi.nlm.nih.gov/ (accessed on 12 November 2025)), Web of Science Core Collection (https://www.webofscience.com (accessed on 12 November 2025)), and KEGG pathway database (https://www.genome.jp/kegg/ (accessed on 12 November 2025)). From the perspective of saprophytic resource insects, two key aspects of the gut of black soldier fly larvae (BSFL; *Hermetia illucens*) are discussed: its potential advantages as a microbial reactor for the degradation of AFB1-contaminated substrates, the microecological mechanism underlying synergistic AFB1 degradation by the core functional flora and the larval host detoxification enzyme system. This review provides a theoretical basis and scientific reference for the application of BSFL intestinal microbiota-mediated AFB1 biodegradation technology in agricultural pollution control, livestock feed detoxification, and the safe treatment of catering kitchen waste.

## 2. Research Progress on AFB1 Detoxification Technology

In current research, detoxification approaches for AFB1 are predominantly categorized into physical, chemical, and biological detoxification. Among them, physical methods enable rapid detoxification but are generally energy-intensive, while chemical methods can disrupt toxin molecular structures yet are accompanied by nutrient degradation and secondary toxic residues [13]. Both approaches share inherent limitations and cannot simultaneously achieve the triple objectives of high efficiency, environmental friendliness, and safety. Concretely speaking, physical detoxification primarily encompasses thermal processing, adsorption, and irradiation methods. However, these approaches have several limitations, including high energy consumption, destruction of feed nutrients, increased risk of feed rancidity, and induction of cytotoxicity [8,14,15,16]. In recent years, photocatalysis and pulsed light technology have emerged, which degrade AFB1 while preserving food quality. However, the stability of photocatalysts and the safety of high-intensity pulsed light in large-scale applications remain to be verified [8]. Chemical detoxification is achieved via the application of chemical agents such as acids, bases, ozone and natural plant extracts to destroy the molecular structure of AFB1, thereby reducing its toxicity [15]. Nonetheless, acid, base or ozone treatments not only adversely affect the nutritional composition and processing properties of feed but also leave substantially harmful chemical residues, leading to secondary pollution [15]. These issues result in significant limitations in practical degradation applications. Natural plant extracts can both degrade AFB1 and alleviate liver toxicity induced by AFB1 exposure. However, their complex composition and cumbersome extraction processes pose challenges to their large-scale promotion and application [17,18,19]. Overall, physical and chemical detoxification methods are plagued by inherent limitations, including high energy consumption, impairment of product quality, and residual toxic substances. Therefore, biological detoxification methods characterized by environmental friendliness, sustainability, and safety have emerged as the focus of subsequent research.

Biological detoxification relies on the adsorption of AFB1 by microbial cell walls and the catalytic degradation of enzymes to reduce the toxicity of AFB1, and it offers distinct advantages such as operating under mild conditions, high specificity, and the absence of toxic residues [13]. However, their degradation efficiency is constrained by the complexity of the treated matrix, which has posed a bottleneck in their large-scale expansion. To date, a multiplicity of microorganisms exhibiting competence in AFB1 detoxification have been isolated from diverse environments, such as soil, animal manure, intestines and water layers (Table 1). Lactic acid bacteria and yeasts are among the microorganisms capable of removing AFB1 via adsorption to their cell walls [20,21]. For example, three strains, *Lactococcus* sp. CF_6, *Lactobacillus* sp. CW_3, and *Lactobacillus acidophilus* CE_4 were isolated from animal excreta, and their AFB1 adsorption rates reached 52.63% to 65.38% [22]. The AFB1 removal ability of beer fermentation residues and five commercial yeast products was assessed, with AFB1 adsorption rates ranging from 24.0% to 69.4% [21]. Although these strains can adsorb AFB1, this adsorption is reversible, and AFB1 will desorb under specific conditions [21,23]. Therefore, the development of microorganisms with effective AFB1-degrading capacity has gradually become the main research direction for AFB1 pollution control. Recent studies have shown that bacteria (e.g., *Bacillus*, *Pseudomonas*, and *Rhodococcus*) and various fungi possess high AFB1 degradation capacity. *Bacillus* plays a vital role in AFB1 pollution control because of its wide distribution and strong stress resistance. For instance, 11 strains of *Bacillus* sp. were isolated from pond silt and soil, and the AFB1 degradation rate of these strains ranged from 27.78% to 79.78% after 48 h [24]. The *Bacillus licheniformis* and *Bacillus subtilis* were isolated from fermented soybeans, which degraded 74% and 85% of AFB1 after 7 days, respectively, and inhibited the growth of *Aspergillus* strains [25]. These studies on AFB1-degrading *Bacillus* provide a theoretical foundation for their applied development. In addition, the combined application of multiple strains achieves better AFB1 removal. For example, *Bacillus* H16v8 and HGD9229 were cocultured, and the degradation rate of the combined strain exhibited 87.7% and 55.3% enhancement, respectively, compared with that of H16v8 and HGD9229 alone [26]. The microbial degradation process typically occurs under mild conditions, without causing severe damage to the nutritional components of food and feed or the production of harmful byproducts that can lead to secondary environmental pollution. This approach meets the requirements of modern society for green and sustainable development and has broad application prospects throughout the human and animal nutrition industries. However, most studies on microbial degradation of AFB1, whether in single or consortia, only employ pure culture-buffer systems; in contrast, practical contaminated matrices (e.g., feed, food, and waste) generally exhibit complex compositions with high lipids and high protein contents, exhibiting significant differences from the former in terms of system complexity and composition [27]. The practical application of these conclusions in real scenarios is therefore limited.

Enzymatic degradation is also one of the common methods for AFB1 detoxification (Table 2). At present, the most widely studied degradation enzymes mainly include laccases (LCs), peroxidases (PODs), and F_420_H_2_-dependent reductases (FDRs). LCs utilize molecular oxygen as an electron acceptor to oxidatively destroy the furan and lactone ring of AFB1, providing an environmentally friendly and highly efficient biodegradation approach [73]. PODs include mainly manganese peroxidase (MnP) and dye-decolorizing peroxidase (DyP). Both enzymes achieve detoxification by oxidizing the double bond in AFB1 to generate AFBO, which is then spontaneously hydrolyzed into the less toxic AFB1-8,9-dihydrodiol [74,75]. FDRs transfer two electrons from F_420_H_2_ to reduce the α,β-unsaturated ester moiety of AFB1, rendering the molecular structure of AFB1 unstable and leading to spontaneous detoxification via hydrolysis [76]. Although microbial degradation enzymes exert a certain effect in AFB1 detoxification, their large-scale application is still limited by the low yield achieved through natural synthesis, high production cost, and poor adaptability of these enzymes to complex processing conditions.

Insects exhibit strong tolerance to specific mycotoxins and grow on mycotoxin-contaminated substrates, converting these toxins into low-toxicity metabolites, thereby achieving mycotoxin detoxification [98]. Therefore, using insect larvae to degrade AFB1 provides a sustainable bioremediation strategy for agriculture and the food industry. Current research has revealed various insect larvae capable of degrading AFB1, such as the navel orange worm (*Amyelois transitella*), codling moth (*Cydia pomonella*), and corn earworm (*Helicoverpa zea*) [99,100]. The larvae of these species can degrade AFB1 into low-toxicity metabolites without producing AFBO. In addition, larvae such as those of the black soldier fly (*Hermetia illucens*), yellow mealworm (*Tenebrio molitor*), and black mealworm (*Alphitobius diaperinus*) achieve an AFB1 degradation rate of more than 50% in the feed matrix while maintaining their growth and survival rates, and they do not accumulate AFB1 in their bodies [101,102,103]. Current research on the biodegradation of AFB1 mostly focuses on detoxifying microorganisms or enzymes in vitro. Although laboratory studies have extensively explored aflatoxin degradation, a mature biological solution for full-scale commercial application remains absent [13]. Various nutrients and inhibitors present in actual contaminated substrates can readily alter the pH, ionic strength, and optimal temperature range of microorganisms or enzymes whether in single or consortia, thereby reducing degradation efficiency [31]. Compared with the use of microbiota or enzymes to degrade AFB1, insects have stronger environmental stress resistance and broader substrate adaptability. They can survive and degrade AFB1 under more diverse, complex, and even harsh environmental conditions and achieve large-scale application through intensive breeding. This approach thus provides an efficient, economical, and sustainable solution to AFB1 pollution.

For BSFL, this degradation ability is attributed to the complex and functionally diverse microbial communities in its intestines, which can serve as microbial reactors for AFB1 degradation. As a case in point, a study experimentally probed the AFB1 degradation ability of BSFL under intestinal sterilized and unsterilized conditions [46]. The results revealed that the AFB1 detoxification efficiencies in sterilized and unsterilized BSFL were 31.71% and 88.72%, respectively, confirming the pivotal function of BSFL intestinal microbiota in AFB1 degradation. In extension of this work, 25 AFB1-degrading strains were isolated from the BSFL intestinal tract, among which *Stenotrophomonas acidaminiphila* A2 exhibited the highest AFB1 degradation capacity, with a degradation rate of 94%. On this basis, the researchers also added a suspension of *Stenotrophomonas acidaminiphila* A2 and BSFL to high-temperature and high-pressure sterilized peanut meal substrate containing 100 ng/g AFB1. The results confirmed that inoculation with this strain mitigated the detrimental effects of AFB1 on BSFL growth performance and enabled BSFL to completely degrade AFB1 within 10 days. These studies unraveled the pivotal function of the intestinal microbiota of insect larvae in AFB1 degradation and opened the possibility for the development of an insect larva-derived intestinal microbiota degradation system. Therefore, BSFLs can not only be employed as living microbial reactors for AFB1 degradation, but their gut-derived microbiota can also be cultivated in vitro as a microbial detoxification agent. However, compared with the traditional large-scale in vitro complex microbial fermentation model, the synergistic system composed of BSFLs and their intestinal microbiota enables efficient and sustaining AFB1 degradation through the integrated metabolism of endogenous enzymes and intestinal bacteria, the regulatory role of immune function in preserving intestinal microbial composition, and the in situ adaptive remodeling of microbiota, thus conferring significant superiority [104,105,106,107]. On the contrary, in vitro cultured complex microbial consortia require specific induction, subculture, or genetic engineering to achieve such an effect, and are susceptible to contamination, leading to substantial and often irreversible declines in degradation efficiency. Moreover, the intestinal lumen of BSFL exhibits an alkaline-weakly acidic-strongly acidic-alkaline pH gradient [108]. Different types of degrading microorganisms and enzymes can continuously degrade AFB1 under this spatial distribution, while it is challenging to replicate such a pH gradient in a single fermentation vessel for in vitro-cultured complex microbial consortia.

## 3. Advantages of BSFL for the Degradation of AFB1 Contaminated Waste

In the process of degrading organic waste, BSFL exhibits broad adaptability and environmental friendliness, making them a highly promising biological treatment approach in the field of organic waste pollution control. For instance, BSFLs are highly effective in degrading various organic solid wastes, including animal manure, food waste, and pharmaceutical industry byproducts [109]. Moreover, the ideal temperature range for BSFL growth and reproduction is 27–37 °C, the optimal relative humidity is 60–70%, and the pH range conducive to larval growth is 6–10 [110]. These growth requirements can be met in many natural and artificial environments. Therefore, BSFLs have strong environmental adaptability for survival under diverse conditions, which further facilitates large-scale breeding. In addition, in the processing of organic waste, BSFLs contribute to the mitigation of greenhouse gas emissions and suppress the abundance of zoonotic pathogens (e.g., *Escherichia coli*, *Salmonella*, and *Staphylococcus aureus*), providing green, low-carbon and eco-friendly environmental benefits [109,111]. In conclusion, BSFLs can adapt to diverse organic waste matrices and conditions and exhibit favorable environmental benefits. Thus, BSFLs hold broad application prospects in the field of organic waste pollution control.

BSFLs exhibit resistance to the most common mycotoxins, such as aflatoxins (AFs), deoxynivalenol (DON), ochratoxin A (OTA), and zearalenone (ZEN) [98]. They can maintain high survival rates and normal growth in feed substrates contaminated with such mycotoxins, significantly reduce AFB1 residues in the substrate, and do not accumulate AFB1 in their bodies. For instance, no significant differences in body weight change or mortality of BSFL between a control group and an experimental group fed feeds supplemented with 4600 μg/kg DON, 260 μg/kg OTA, 88 μg/kg AFB1, 17 μg/kg AFB2, 46 μg/kg AFG2 and 860 μg/kg ZEN [112]. A 97.3% survival rate of BSFL was reported when the larvae were fed a wheat-based diet spiked with 0.5 mg/kg AFB1 [113]. BSFL effectively degraded 83–95.1% of AFB1 while maintaining a high survival rate in a feed matrix supplemented with 0.415 mg/kg AFB1, and the AFB1 concentrations in the freeze-dried larvae fell below the analytical detection threshold (<0.10 µg/kg) [101]. Another study found no statistically significant disparities in the average survival rate (94–100%) or average larval fresh weight (172–191 mg/larva) of BSFL between the control group and the groups administered with 8–430 μg/kg AFB1, 170–2000 μg/kg OTA, 280–13,000 μg/kg ZEN, 3900–112,000 μg/kg DON, or a mixture of mycotoxins [114]. Targeted mass balance calculations were performed for AFB1 degradation. The results revealed that the average total mass balance recovery rates of AFB1 and its metabolites in BSFL treatment groups ranged from 11 to 18% [114]. Among these metabolites, only aflatoxicol (AFL) was detected, accounting for 0.2% of the total mass balance. The low residual levels of AFB1 and its metabolites indicate that BSFLs have an extremely strong degradation capacity for AFB1. Across all treatment groups, AFB1 residues in BSFL were undetectable (i.e., below the methodological detection limit), and while DON, OTA, and ZEN were present, their concentrations were significantly lower than those in residual feed [114]. Thus, at the abovementioned experimental concentrations, BSFL efficiently degraded AFB1 while maintaining normal growth and survival, whether in substrates contaminated with AFB1 or mycotoxin mixtures. This dual capacity provides reliable support for the biological detoxification of AFB1 in practical complex contamination scenarios.

In conclusion, BSFLs have demonstrated the core application advantage of integrating feasibility and efficiency in the practical treatment of AFB1 contamination. From the perspective of application feasibility, BSFLs have broad adaptability to diversified organic residues, and conditions suitable for their growth are easily achieved. This comprehensive adaptability lays a foundation for their large-scale cultivation and practical application. In terms of AFB1 detoxification efficacy, BSFLs can maintain normal growth and development in substrates contaminated with mycotoxins and efficiently degrade AFB1 without residues in their bodies, thereby avoiding the risk of secondary accumulation of toxins in food chains. These characteristics of BSFL provide practical and safe technical support for their use in detoxifying AFB1 contamination in complex organic waste matrices.

## 4. Composition, Source, and Colonization of BSFL Intestinal Microbiota

The intestinal tract of insects harbors an intricate microbiota consisting of protozoa, fungi, bacteria, and archaea, among which bacteria are the dominant species [115]. A variety of bacterial taxa in the intestinal tract of BSFL have been identified to date (Table 3). Among them, Actinobacteria, Bacteroidetes, Firmicutes, and Proteobacteria are the dominant bacterial phyla. The genera mainly include *Pseudomonas*, *Enterococcus*, *Providencia*, *Escherichia*, *Klebsiella*, *Enterobacteriaceae*, *Stenotrophomonas*, *Acinetobacter*, *Dysgonomonas*, and *Morganella*. These genus-level taxa are considered potential core members of the BSFL intestinal microbiota and play important roles in nutritional metabolism, organic matter degradation, and immune regulation. While most studies on the composition of BSFL intestinal microbiota focus on bacterial communities, some studies have identified fungi as a major component. In existing studies, the fungal species in the intestinal tract of BSFL mainly belong to the phylum Ascomycota, including the genera *Pichia*, *Candida*, *Diutina*, *Cyberlindnera*, *Aspergillus*, *Geotrichum*, and *Trichosporon* [116,117,118,119,120]. Although the abovementioned microorganisms are ubiquitous, no current research has confirmed that a specific microbial group maintains absolute dominance or persistently high abundance across all environmental conditions. Thus, despite the presence of some dominant groups, the intestinal microbiota of BSFL exhibits strong flexibility and autonomous adaptability.

There are two main sources of intestinal microbiota in BSFL. One is the endogenous microbiota carried by larvae upon hatching [120]. The microbiota present on the body surface and in the intestinal tract of adult flies is vertically transmitted from the maternal parent to progeny through oviposition, becoming the initial component of the larval intestinal microbiota at hatching. The second is the exogenous microbiota acquired by larvae from the environment through feeding [120]. The core microbiota vertically transmitted from the mother preferentially colonizes the intestinal tract immediately after larval hatching, forming an initial microbiota structure and providing a basic framework for the subsequent enrichment of environmental microorganisms. Environmentally derived microorganisms are the core input source of the intestinal microbiota of BSFL. When BSFLs intake moldy substrates contaminated with AFB1, they also take in naturally occurring functional microorganisms in these substrates, such as potential AFB1-degrading bacteria, including *Bacillus*, *Lactobacillus*, and *Pseudomonas*. Thus, an intestinal functional community centered on AFB1-degrading bacteria gradually develops, providing a stable foundation for the subsequent synergistic degradation of AFB1 by the host and intestinal microbiota.

The colonization of intestinal microbiota in BSFL is regulated by the intestinal structure and immune function of the larva. In terms of intestinal structure, the BSFL intestine is mainly divided into the foregut, midgut, and hindgut. The midgut compartment is further subdivided into the anterior midgut (AMG), middle midgut (MMG), and posterior midgut (PMG) [108]. The pH in the BSFL midgut lumen presents a three-level gradient of “weakly acidic (pH = 5.9)-strongly acidic (pH = 2.1)-alkaline (pH = 8.3)”. This pH gradient acts as a chemical barrier that aids in pathogenic microorganism elimination and nutrient recovery, thereby maintaining the stability of the BSFL intestinal flora [108]. The MMG region is a key microbial screening site in the BSFL intestine. The strongly acidic environment and activity of lysozymes in the MMG region strongly affect the distribution of intestinal bacteria [105]. Under selective pressure in the MMG region, the community diversity of the BSFL microbiota from the AMG to the hindgut significantly decreases from high to low, whereas the bacterial load increases [161]. Moreover, the folds and microvilli on the BSFL intestinal wall also provide abundant attachment sites for microorganisms, enhancing their colonization capacity [115]. Host immune function is another factor regulating the colonization of BSFL intestinal flora. BSFLs regulate the synthesis and secretion of humoral immunity-related effector molecules (e.g., antimicrobial peptides and lysozyme), which mediate immune responses to screen intestinal microorganisms, thus effectively inhibiting the colonization of pathogenic microorganisms and maintaining the compositional and functional stability of beneficial gut microbes [105,162]. It can be concluded that the colonization of intestinal microbiota in BSFL is jointly determined by the environmental conditions provided by their intestinal structure and the immune filtering regulated by the larval host.

In conclusion, the initial sources and subsequent colonization screening of the BSFL intestinal microbiota jointly determine its community stability, which constitutes the core foundation for establishing the synergistic degradation function of AFB1. At the source level, the BSFL intestinal microbiota comprises vertically transmitted endogenous initial microbiota and exogenous functional microorganisms acquired by larvae via feeding on moldy substrates, thereby forming a functional microbiota centered on AFB1 degradation. At the colonization level, this process depends on the synergistic effect of BSFL intestinal structure and immune function, which ensures the stable colonization of intestinal microorganisms, laying a solid foundation for the host–microbiota collaborative detoxification of AFB1. Therefore, clarifying the sources and key regulatory factors of the intestinal microbiota of BSFL is highly important for investigating the synergistic interactions between BSFL and its intestinal microbiota.

## 5. Synergistic Degradation of AFB1 by Endogenous Enzymes and Intestinal Microbiota of BSFL

BSFLs possess a variety of enzymes involved in exogenous compound biotransformation, which catalyze the conversion of plant secondary metabolites, pesticides or mycotoxins into more polar water-soluble metabolites for excretion [163]. The detoxification enzyme system of BSFL, which targets AFB1, also operates via this pathway. The metabolism of AFB1 can be divided into three phases (Figure 2A). In Phase I metabolism, AFB1 is converted mainly into more polar metabolites via oxidation, reduction, and hydrolysis reactions [104]. The CYP450 enzyme of BSFL catalyzes the conversion of AFB1 into aflatoxin P1 (AFP1) and AFBO, while cytoplasmic NADPH-dependent reductase (NPR) mediates the conversion of AFB1 to AFL [104,113]. AFBO is converted to AFB1-dihydrodiol by juvenile hormone epoxide hydrolase 1 (JHEH), which further undergoes spontaneous rearrangement to form AFB1-dialdehyde [104]. This dialdehyde is subsequently converted to AFB1-dialcohol by aldo-keto reductase family 1 member B1 (AKR1B1) [104]. Phase II metabolism encompasses the coupling of intermediate metabolites of AFB1 (yielded in Phase I) with endogenous hydrophilic substances, resulting in the production of metabolites with diminished toxicity, elevated polarity, and increased aqueous solubility [104]. AFBO undergoes molecular conjugation with glutathione (GSH) via glutathione-S-transferase (GST)-facilitated enzymatic catalysis, culminating in the formation of the AFB1-glutathione conjugate (AFB1-GSH) [104]. Under the catalysis of γ-glutamyl transferase (GGT), dipeptidase (DPEP), and N-acetyltransferase (NAT), AFB1-GSH is further metabolized into AFB1-mercapturic acid and eventually excreted [164]. AFB1-dialcohol conjugates with glucuronic acid under the catalysis of UDP-glucosyltransferase 2 (UGT-2) to form AFB1-glucuronide, which is then excreted [104,164]. Phase III metabolism involves the efflux via membrane transporters of the hydrophilic conjugates generated in Phases I and II [104]. The hydrophilic conjugates formed by AFB1 metabolism in BSFL are excreted from cells via the transmembrane transporter multidrug resistance protein 2 (MRP2), thereby protecting BSFL from AFB1 toxicity [104]. The detoxification enzyme system of BSFL catalyzes the formation of less toxic metabolites (e.g., AFL, AFP1, AFB1-dihydrodiol) from AFB1, which are then transported to the intestinal lumen. In conjunction with the robust metabolic capacity of the BSFL intestinal microbiota, complete degradation of AFB1 is achieved (Figure 2B). Thus, the three-phase metabolic pathway facilitates metabolic synergy between the endogenous detoxification enzyme system of BSFLs and their intestinal microbiota, ultimately enabling the joint completion of AFB1 biotransformation and degradation processes.

The specific degrading flora in the intestine of BSFL is the key driver of AFB1 degradation. For instance, *Stenotrophomonas acidaminiphila* A2 was screened from the intestinal tract of BSFL and reported a 94% AFB1 degradation efficiency [46]. In addition, in the studies on environmental AFB1-degrading bacteria mentioned above, *Bacillus* and *Pseudomonas* are widely reported and common members of the BSFL intestinal microbiota. Thus, they are presumed to play a key role in AFB1 degradation by BSFL. *Bacillus* yields a variety of degrading enzymes, including DyP peroxidase, CotA laccase, N-acyl homoserine lactone-degrading enzyme (AHL-lactonase), α/β hydrolase (ABH), aldo/keto reductase (AKR), and bacilysin biosynthesis oxidoreductase (BacC) [24,33,34,90,97]. CotA and BacC are aerobic enzymes, and their activity is low in the central anoxic lumen (≤2.5% O_2_). On the contrary, ABH and AHL-lactonase (hydrolytic), DyP peroxidase (uses trace H_2_O_2_), and AKR (NADPH-dependent reduction), all of which retain high AFB1-degrading capacity under anoxic conditions [24,34,87,94,95,96]. These enzymes directly target the active site of AFB1 and degrade it into low-toxicity or nontoxic metabolites. In addition, *Bacillus* species synthesize various lipopeptide antibiotics (e.g., surfactin, fengycin, and bacillomycin D), disrupting the cell membrane integrity of *Aspergillus flavus* and thereby inhibiting its growth and reducing AFB1 biosynthesis [166]. *Pseudomonas* species mediate the biodegradation of aromatic compounds (e.g., benzene, toluene, and xylene) via enzyme systems associated with lipid degradation (e.g., lipase and β-oxidase) and alkane oxidation (e.g., various monooxygenases) [42,167,168,169]. As an aromatic compound, AFB1 is presumed to be degraded through a similar enzyme-mediated catabolic pathway [42]. In addition, strains of *Myroides*, *Escherichia*, *Staphylococcus*, *Lysinibacillus*, *Enterobacter*, *Klebsiella*, *Aspergillus*, *Microbacterium*, and *Enterococcus* have been demonstrated to harbor AFB1-detoxifying capacities (Table 1), and these genera are frequently detected in intestinal microbiota analyses of BSFL (Table 3). Considering the anoxic environment of the BSFL intestinal tract, facultative anaerobic bacteria including *Bacillus*, *Pseudomonas*, *Escherichia*, *Stenotrophomonas*, *Enterococcus*, *Staphylococcus*, *Enterobacter*, and *Klebsiella* can survive and exert metabolic activities in the anoxic intestinal segments of BSFL [170,171,172,173,174,175,176,177]. Therefore, they may directly participate in AFB1 degradation mediated by the BSFL intestinal microbiota. On the contrary, *Aspergillus*, *Myroides*, *Lysinibacillus*, and *Microbacterium* are aerobes [178,179,180,181]. Their mycelial growth or respiratory chains require continuous oxygen supply, rendering them unable to perform active oxidative metabolism of AFB1 in the anoxic central segments of the intestinal lumen.

BSFL initiates more efficient degradation pathways by specifically reshaping the composition of their intestinal microbiota and enhancing the expression of degradation enzyme-encoding genes in response to the survival pressure imposed by environmental pollutants [106,107]. Therefore, under AFB1 stress, microorganisms with AFB1 degradation capacity become dominant in the BSFL intestinal microbiota because they replace the original dominant flora and highly express key AFB1-degrading enzymes such as DyP, AHL-lactonase, ABH and AKR [24,90,97]. These enzymes target the coumarin lactone ring and difuran ring of AFB1 through oxidation, reduction and hydrolysis reactions, affecting the transformation of highly toxic AFB1 molecules into metabolites with significantly reduced or even abolished toxicity, which are readily eliminated by BSFL or the surrounding environment (Figure 2C). BSFL and their intestinal microbiota respond flexibly to pollution stress through a three-level synergy of “dynamic remodeling of community structure–directional regulation of gene expression–enhancement of enzyme activity”. This close interaction plays an important role in promoting the degradation of AFB1. In addition to direct degradation, lactic acid bacteria (e.g., *Lactobacillus*, *Weissella*, and *Pediococcus*) and yeast (e.g., *Pichia*, *Candida*, *Geotrichum*, and *Trichosporon*) adsorb AFB1 via components such as peptidoglycan, teichoic acid, and β-glucan in their cell walls [20,21,56,156,182,183,184,185,186,187] (Figure 2D). *Lactobacillus*, *Weissella*, *Pediococcus*, *Pichia*, *Candida*, *Geotrichum*, and *Trichosporon* are all facultative anaerobes that can survive and exert adsorptive activity in the anoxic intestinal tract of BSFL, thereby reducing the bioavailability of AFB1 in the intestinal tract and affording protection against injury to the host [87,188,189,190,191,192]. Furthermore, lactic acid bacteria can also potentiate the nuclear factor erythroid 2-related factor 2 (Nrf2) pathway [193]. On the one hand, these species increase the activity of antioxidant enzymes and reduce lipid peroxidation levels, alleviating AFB1-induced oxidative stress. On the other hand, they upregulate the expression of GST, thereby enhancing the body’s ability to detoxify AFBO. The initiation and progression of AFB1 degradation by the BSFL intestinal microbiota relies on the secretion of degradation enzymes by core microorganisms and the auxiliary role of other microorganisms in the community. These two aspects jointly sustain the efficient progression of the degradation process.

## 6. Possible Pathways of AFB1 Degradation by the Intestinal Microbiota in BSFL

The current research on the biodegradation of AFB1 mainly focuses on redox reactions as the dominant mechanism. The unsaturated sites on the furan ring, lactone ring, and cyclopentenone ring of AFB1 undergo oxidation, reduction, or hydroxylation reactions, which block the binding ability of AFB1 to DNA, thereby reducing its carcinogenic risk and forming metabolites with low stability. Among the main pathways for AFB1 biodegradation is an oxygenation reaction to yield highly toxic AFBO, which is subsequently converted into the less toxic AFB1-dihydrodiol [73,74]. This step can be catalyzed by the endogenous detoxification enzyme system of BSFL. *Bacillus subtilis*, *Bacillus licheniformis*, and *Bacillus amyloliquefaciens* convert AFB1 into its hydroxyl derivative Aflatoxin Q1 (AFQ1) via the secretion of laccase, whose toxicity and mutagenicity are 1/18 and 1% of those of AFB1, respectively [77,81,194,195,196]. *Aspergillus niger* not only reduces the ketone carbonyl group on the cyclopentenone ring of AFB1 to form the AFL, but also mediates the reductive elimination of AFB1 after hydrolysis and ring opening of its lactone ring [197,198]. The BacC enzyme secreted by *Bacillus subtilis* can reduce the α,β-unsaturated ester moiety of AFB1 [97]. After these reactions, the key toxic sites of AFB1 are destroyed, resulting in a significant reduction in its carcinogenicity and mutagenicity. However, in the microaerophilic or anaerobic intestinal environment of BSFL, oxidative reactions exhibit low activity, and the degradation of AFB1 primarily relies on non-oxidation catalyzed reactions mediated by the intestinal anaerobic or facultative anaerobic microbial community.

In the intestinal tract of BSFL, facultative anaerobes, such as *Bacillus*, *Pseudomonas*, *Escherichia*, *Stenotrophomonas*, and *Enterococcus*, can degrade AFB1 by non-oxidative reactions [41,46,57,90,118,132,197]. On the basis of the AFB1 degradation mechanisms of these microorganisms, the possible degradation pathways of the intestinal microbiota in BSFL can be preliminarily inferred (Figure 3). *Bacillus subtilis* not only catalyzes reactions to yield AFBO (P-1) and AFB1-dihydrodiol (P-2) by secreting dye-decolorizing peroxidase (BsDyP) but also catalyzes hydration reactions to form product P-3 [90,199]. *Enterococcus faecium* can reduce the α,β-unsaturated ester moiety of AFB1(P-4), causing it to undergo spontaneous hydrolysis [57]. Apart from these minor redox reactions, hydrolysis and cracking reactions play a dominant role. After hydrolysis and cracking reactions, further reactions such as demethylation, demethoxylation, or decarbonylation gradually cleave it into various low-toxicity or nontoxic small-molecule products, thereby reducing the residue and toxicity of AFB1 in the environment [46,199]. *Pseudomonas putida* catalyzes the cleavage of the lactone ring of AFB1, and the cleaved ring undergoes hydrolysis and decarbonylation to produce Aflatoxin D1 (AFD1, P-5) [41]. Further reactions generate Aflatoxin D2 (P-6) and Aflatoxin D3 (phthalic anhydride, P-7), whose lactone ring and cyclopentenone ring are lost [41]. The molecular formula of the metabolite generated from the degradation of AFB1 by *Escherichia coli* is C_16_H_14_O_5_, which is tentatively identified as AFD1 or its isomers [64]. In a *Bacillus* coculture degradation system, AFB1 is converted into AFD1 [26]. Furthermore, the hydroxyl group is added to the dihydrofuran ring and the benzene ring side chain to generate product P-8 [26]. Subsequently, two CO groups are removed from the furan ring to produce product P-9, and P-9 undergoes the removal of one methyl group from its benzene ring side chain to yield product P-10 [26]. *Stenotrophomonas acidaminiphila* A2 removes one carbonyl group from the furan ring of AFB1 and one methyl group from the coumarin ring, yielding product P-11 [46]. *Bacillus subtilis* removes the methoxy group from AFB1 to generate product P-12 and simultaneously generates a series of small-molecule compounds (P-13 to P-17) in which the furan ring and coumarin structure of AFB1 are destroyed [199]. *Enterococcus faecium* hydrolyzes the lactone ring of AFB1 to form product P-18, and the hydrolyzed lactone ring then undergoes decarboxylation and cleavage to form product P-5, which is further converted into product P-19 via furan ring removal [57]. *Bacillus albus* also targets the difuran ring and lactone ring as its main degradation sites, thereby generating products P-4, P-5, P-6, and P-20 to P-21 [200]. Moreover, molecular polarity is markedly enhanced, facilitating the transformation and excretion of AFB1 from the body via the metabolic system, thus achieving the conversion from a highly toxic substance to low-toxicity and easily removable products.

On the basis of the above analysis of the key enzymatic catalytic sites and product characteristics of AFB1, the AFB1 degradation pathway mediated by the intestinal microbiota of BSFL is summarized herein. First, AFB1 undergoes oxidation, reduction, or hydroxylation reactions, which specifically modify the unsaturated bonds of the furan ring, lactone ring, and cyclopentenone ring in its molecule, thereby reducing its carcinogenic risk. The lactone ring and furan ring of AFB1 subsequently undergo further hydrolytic reactions. The active sites exposed after ring opening can continue to undergo reactions such as demethylation, demethoxylation or decarbonylation. These reactions gradually destroy the molecular skeleton of AFB1, eventually generating various low-molecular-weight metabolites. The central pathway for the biotransformation of aromatic compounds ends with the citric acid (TCA) cycle [47]. The various intermediate metabolites formed by AFB1 degradation may be completely oxidized after they participate in the TCA cycle and eventually be converted into harmless small molecules such as CO_2_ and H_2_O or assimilated into BSFL biomass, completely eliminating the harm caused by AFB1 to organisms.

## 7. Conclusions and Prospects

Research on the degradation of AFB1 by BSFL has opened a novel path for solving the problem of AFB1 pollution. Conventional physicochemical remediation strategies for AFB1 present numerous constraints, including high energy consumption, high resource consumption, high cost, and the propensity to generate toxic byproducts. Therefore, biodegradation methods, especially those using the intestinal tracts of insect larvae as microbial reactors for AFB1 degradation, have gradually emerged, providing new ideas for the safe and effective degradation of AFB1. BSFLs possess the dual advantages of broad-spectrum mycotoxin tolerance and a highly efficient detoxification capability. This species can tolerate and handle the coexistence of multiple mycotoxins commonly found in contaminated substrates. Through the synergistic effect of its detoxification enzyme system and intestinal microorganisms, it gradually metabolizes AFB1 into low-toxicity or nontoxic products. Therefore, BSFLs have promising industrial application prospects. Evidence derived from the aforementioned analyses indicates that the detoxification of AFB1-contaminated substrates by BSFL intestinal microbiota can provide a new approach for ensuring food safety, promoting the green and efficient control of AFB1 pollution, and facilitating the sustainable utilization of resources.

Recent progress has been made in the research on AFB1 degradation mediated by the intestinal microbiota of BSFL, including AFB1 concentration tolerance, degradation efficiency, and screening of functional degrading bacteria. However, many bottlenecks still need to be addressed in terms of systematic analyses of the mechanisms involved, optimization of the degradation efficiency and expansion of the scenarios in which BSFLs are applicable for AFB1 degradation. To address the core issues of which microorganisms or genes mediate AFB1 degradation, metagenomic, macro-transcriptome and metabolomic technologies can be combined. The core microorganisms and potential AFB1 degradation genes or gene clusters involved in degradation can be explored by comparing the differences in microbial community composition, gene expression and metabolites between the AFB1 stress group and the control group. The metabolic flow of degradation intermediate products can be tracked, and ultimately, the complete AFB1 biodegradation pathway can be mapped. Further, single strains can be isolated and screened from the identified core degradation microflora to identify the best AFB1-degrading strains. The key degradation enzyme-encoding genes can be heterologously expressed, purified and verified to obtain the target degradation enzymes. With respect to the degradation strains and enzymes obtained from the intestinal microbiota screening of BSFL, genome editing can be adopted to directionally optimize the metabolic pathways of the strains, or protein engineering can be used to modify the key functional domains of the degradation enzymes, thereby increasing strain and enzyme stability and degradation efficiency in the complex feed-processing environment. Meanwhile, the detoxification capacity of intact BSFL can be further enhanced by modulating their intestinal microbiota. For instance, the inoculation of AFB1-degrading strains isolated from the BSFL gut into the rearing substrate or the enrichment of indigenous degrading bacteria through dietary manipulation serve as effective strategies. In vitro cytotoxicity tests and animal model feeding tests can be performed to systematically detect the acute toxicity and genotoxicity of AFB1 degradation products and their impacts on the growth performance of livestock and poultry. Moreover, it is necessary to verify the stability of the BSFL-mediated degradation model under large-scale rearing conditions. Key indicators such as the degradation rate, product residue, and microbial safety limits can be clearly defined to establish a large-scale, standardized biodegradation process and a complete risk prevention and control system, thereby providing a viable pathway for the application of BSFL in organic waste treatment.

The current application of BSFL for AFB1 degradation still faces notable legislative limitations, particularly the lack of targeted regulatory standards for its application in AFB1-contaminated feed or food-related matrices and the ambiguity in legal liability for product safety. For instance, the European Union (EU) and the American Association of Feed Control Officers (AAFCO) legislation impose strict regulations on the types, sources and application of substrates for insect farming, explicitly requiring the use of feed-grade raw materials from reliable sources [201,202,203,204]. Therefore, this regulatory requirement directly excludes mycotoxin-contaminated substrates from the scope of legally permissible substrates for BSFL cultivation, which are precisely the toxic substrates targeted by the “insect-mediated biotransformation of toxic substrates” technology [201,202]. Consequently, this technology lacks a feasible and compliant implementation pathway within the existing regulatory framework. Even though such waste can be degraded and recycled by insects, their large-scale application in insect farming remains prohibited due to legislative restrictions. In the future, it is essential to advance the coordinated alignment between legislation and technical practices. Specifically, a feasible pathway for the large-scale application of BSFL-mediated AFB1 degradation can be established by defining safety standards and application boundaries for contaminated substrates.

## Figures and Tables

**Figure 1 animals-15-03351-f001:**
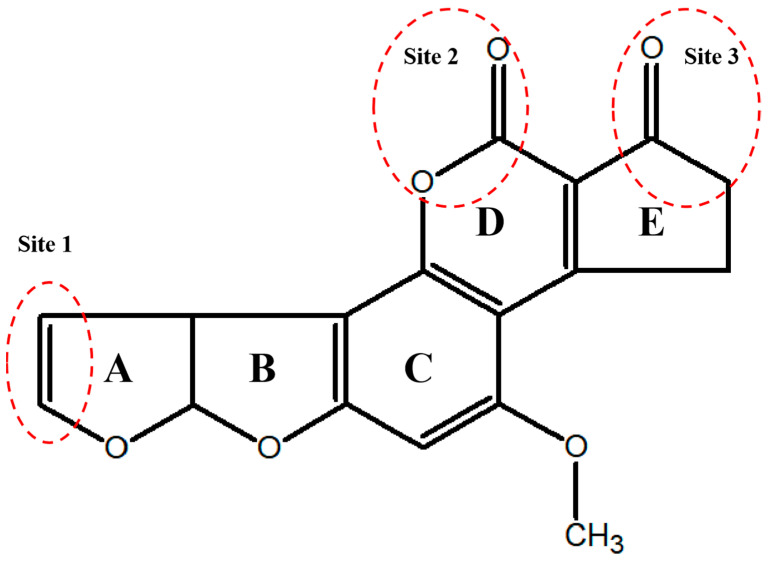
AFB1 structure and its active sites include a dibenzofuran ring that is composed of ring A (furan ring) and ring B (cyclohexane ring), a coumarin skeleton containing ring C (naphthalene ring) and ring D (lactone ring), and a cyclopentenone ring (ring E). Site 1 undergoes epoxidation to form the toxic effector AFBO, which induces genotoxicity and carcinogenicity. Site 2 is the active site where AFB1 undergoes hydrolysis. Differences in the substituents on Site 3 affect the toxicity of AFB1.

**Figure 2 animals-15-03351-f002:**
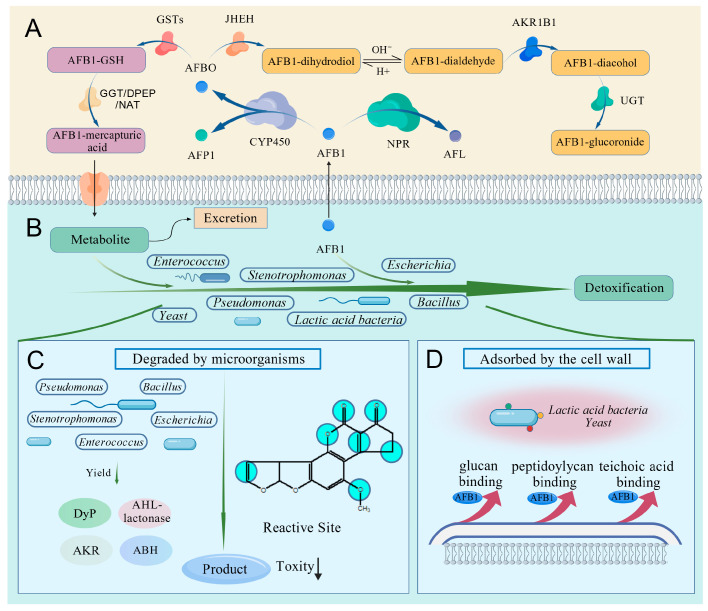
(**A**) AFB1 degrading by the BSFL detoxifying enzymes system; (**B**) Synergistic AFB1 degradation by the BSFL intestinal microorganisms; (**C**) Gut microbes can secrete certain extracellular enzymes to destroy specific structural sites of AFB1, including lactone rings, furan ring double bonds, and cyclopentanone carbonyl groups, achieving the effect of reducing the toxicity of AFB1; (**D**) The intestinal microorganisms of BSFL adsorb AFB1 by cell wall components such as peptidoglycans, polysaccharides, and phosphate groups, thereby reducing the bioavailability of AFB1 in the intestine [165]. Denote: CYP450, Cytochrome P450; NPR, NADPH-dependent reductase; JHEH, Juvenile hormone epoxide hydrolase 1; AKR1B1, Aldo-keto reductase family 1 member B1; UGT, UDP-glucosyltransferase; GSTs, Glutathione S-transferases; GSH, Glutathione; GGT, γ-glutamyl transpeptidase; DPEP, Dipeptidase; NAT, N-Acetyltransferase; DyP, Dye-Decolorizing Peroxidase; AHL-lactonase, N-acyl-homoserine lactonase; ABH, α/β hydrolase; AKR, aldo/keto reductase. Lactic acid bacteria: *Lactobacillus*, *Weissella*, and *Pediococcus*; Yeast: yeast functional group includes both true yeasts and yeast-like fungi that can adsorb AFB1, such as *Pichia*, *Candida*, *Geotrichum*, and *Trichosporon*.

**Figure 3 animals-15-03351-f003:**
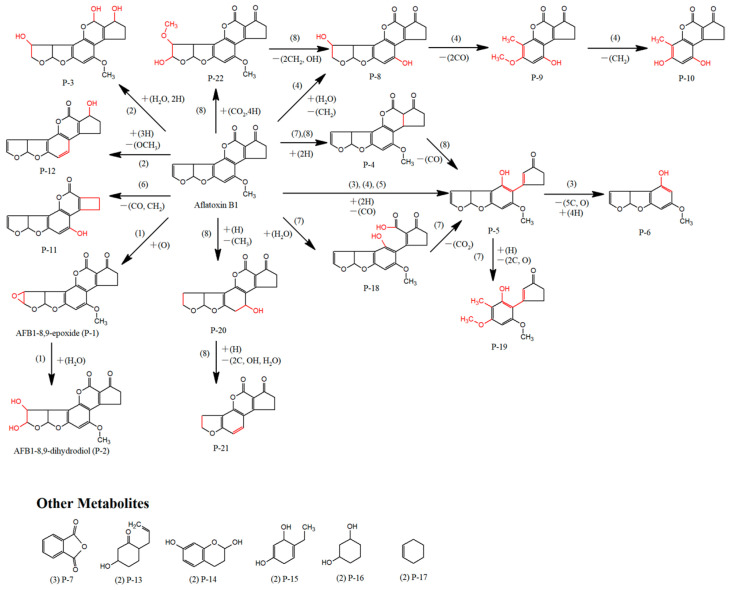
Potential degradation pathways of AFB1 by BSFL intestinal microorganisms. Denote: (1) *Bacillus subtilis* (BsDyP). (2) *Bacillus subtilis* (unknown). (3) *Pseudomonas putida* (unknown). (4) *Bacillus* sp. (unknown). (5) *Escherichia coli* (unknown). (6) *Stenotrophomonas acidaminiphila* (unknown). (7) *Enterococcus faecium* (unknown). (8) *Bacillus albus* (unknown). The red mark in the figure denotes the reaction site.

**Table 1 animals-15-03351-t001:** Microorganisms that can be used for the detoxification of AFB1 and their degradation effects.

Class	Source	Name	AFB1 Concentration	Incubation Period	Reduction Efficacy (%)	References
*Bacillus*	Maize Grains	*Bacillus* sp. TUBF1	10 μg/mL	48 h/72 h	81.50/100.00	[28]
	Thua-nao	*B. licheniformis*, *B. subtilis*	5 mg/L	7 d	74.00/85.00	[25]
	/	*B. licheniformis* CFR1	500 μg/kg	72 h	94.70	[29]
	Pistachio nuts	*B. subtilis* UTBSP1	2 μg/kg	5 d	95.00	[30]
	Hog deer feces and farm soil	*Bacillus* sp.	100 μg/kg	72 h	77.80–80.93	[31]
	Plant leaf	*B. aryabhattai*	2 μg/mL	72 h	82.92	[32]
	Pond mud and soil	*Bacillus strains* (11)	500 ng/mL	48 h	27.78–79.78	[24]
	Rotten feed	*B. subtilis* WJ6	5 μg/mL	48 h	81.57	[33]
	/	*B. halotolerans* DDC-4	1 μg/mL	72 h	76.30	[34]
	/	*B. subtilis*	40 μg/L	24 h	38.38	[35]
	Polygalae	*B. megaterium* SX1-1	0.1 ng/mL	72 h	97.45	[36]
	Qinghai–Tibet Plateau	*B. amyloliquefaciens* YUAD7	10 μg/mL	72 h	91.70	[37]
	Sitophilus oryzae gut	*B. subtilis* RWBG1, *B. oceanisediminis* RWGB2, *B. firmus* RWGB3	1 μg/kg	48 h	63.60–84.20	[38]
*Pseudomonas*	Gold mine aquifer	*P. anguilliseptica* VGF1	5 μg/kg	48 h	51.70	[39]
		*P. fluorescens*			47.70	
	Peanut-growing soils	*P. knackmussii* AD02	100 ng/mL	24 h	90.00	[40]
	/	*P. putida* MTCC 1274 and 2445	0.2 μg/mL	24 h	90.00	[41]
	Sitophilus oryzae gut	*P. aeruginosa* RWGB4	5 μg/kg	48 h	48.90	[38]
	Farm soils, maize and rice	*P. aeruginosa* N17-1	100 μg/kg	72 h	82.80	[42]
	Hydrocarbon-contaminated sites	*Pseudomonas* sp. (6)	2 μg/mL	72 h	80.14–97.61	[43]
*Stenotrophomonas*	South American tapir feces	*Stenotrophomonas maltophilia* 35-3	100 μg/kg	72 h	82.50	[31]
	/	*Stenotrophomonas* sp. NMO-3	100 μg/kg	72 h	85.70	[44]
	/	*Stenotrophomonas* sp. CW117	45 μg/L	24 h	100.00	[45]
	Black soldier fly larval gut	*Stenotrophomonas acidaminiphila* A2	0.1 μg/mL	48 h	94.00	[46]
*Rhodococcus*	Ostrich feces	*Rhodococcus* sp.	100 μg/kg	72 h	73.92	[31]
	/	*R. erythropolis* ATTC 4277	20 μg/mL	24 h	96.00	[47]
	Polycyclic aromatic hydrocarbons contaminated soils	*R. erythropolis*	1.75 mg/kg	72 h	66.80	[48]
	Hydrocarbon-contaminated sites	*Rhodococcus* sp. (16)	2 μg/mL	72 h	20.79–99.98	[43]
	Oil-contaminated soil and Natural soil	*Rhodococcus* sp. (32)	2 mg/kg	72 h	20.00–100.00	[49]
	Soil	*R. pyridinivorans* 4-4	0.1 μg/mL	24 h	84.90	[50]
	/	*R. turbidus* PD630	0.4 μg/mL	72 h	93.04	[51]
	/	*Rhodococcus* Strains (42)	3 μg/mL	3 d	17.00–100.00	[52]
	Polycyclic aromatic hydrocarbons contaminated soils	*R. erythropolis* DSM 14303	1.75 mg/kg	72 h	94.00–97.00	[53]
Lactic acid bacteria	/	*Lactobacillus plantarum* PTCC 1058	240 mg/kg	4–7 d	77.00	[54]
	/	*Lactobacillus casein*	40 μg/L	24 h	26.06	[35]
	Fermented foods	*Lactobacillus plantarum*	150 μg/L	24 h	89.50	[55]
	/	*Levilactobacillus brevis* (2), *Lactobacillus helveticus* (9), *Lactoplantibacillus plantarum* (4)*, Leuconostoc* sp. (4)*, Pediococcus claussenii*, *Weissella* sp. (9)	1 μg/mL	1.5 h	Binding: 16.10–40.90 (viable)/29.60–65.70 (non-viable)	[56]
	Grassland soil	*Enterococcus faecium* HB2-2	105.1 µg/kg	96 h	82.90	[57]
Other Bacterial Genera	Polycyclic aromatic hydrocarbon-contaminated soils	*Mycobacterium fluoranthenivorans* sp. *nov.* DSM44556T	1.75 mg/kg	24 h	~100.00	[53]
	Contaminated soil of a former coal gas plant	*Mycobacterium fluoranthenivorans* FA4T	2.5 mg/kg	72 h	Leaving no detectable AFB1	[58]
	Soil	*Streptomyces* (59)	2 μg/mL	5 d	43.00–94.00	[59]
	/	*Str. lividans* TK 24, *Str. aureofaciens* ATCC 10762	20 μg/mL	24 h	88.00/86.00	[47]
	/	*Streptomyces* sp.	1 mg/L	5 d	88.40	[60]
	Hydrocarbon-contaminated sites	*Streptomyces* sp. (2), *Arthrobacter protophormiae*, *Microbacterium* sp. (2), *Pseudoxanthomonas* sp. (2), *Chryseobacterium* sp. (2)	2 μg/mL	72 h	56.88–80.22	[43]
	Gold mine aquifer	*Lysinibacillus fusisormis*, *Sporosarcina* sp., *Staphylococcus warneri*	2.5 μg/mL	48 h	46.90–61.30	[61]
	/	*Staphylococcus* sp. VGF2	5 μg/kg	48 h	56.80	[39]
	/	*Flavobacterium aurantiacum* NRRL B-184	10 μg/mL	48 h	81.10	[62]
	Corn-planted soil	*Brevundimonas* sp. LF-1, *Brevundimonas* sp. (2), *Brachybacterium* sp., *Klebsiella* sp., *Enterobacter* sp., *Cellulosimicrobium* sp.	2 mg/L	72 h	86.90	[63]
	Animal feces, farm soil		100 μg/kg	72 h	73.75–78.10	[31]
	Chicken Cecum	*Escherichia coli* CG1061	2.5 μg/mL	72 h	93.70	[64]
	/	*Myroides odoratimimus* Strain 3J2MO	100 μg/kg	48 h	93.82	[65]
Mixed strains	/	*Bacillus subtilis*, *Lactobacillus casein*, *Candida utilis*	40 μg/L	24 h	45.49	[35]
	/	*Streptococcus thermophilus* and *Lactobacillus delbrueckii* subsp. *bulgaricus*	10.5 μg/kg	3 d	100.00	[66]
Fungi	Fermented soybean	*Aspergillus niger* FS10	1 μg/mL	72 h	98.65	[67]
	/	*Aspergillus flavus*	5 mg/kg	4 d	0–84.40	[68]
	/	*Candida utilis*	40 μg/L	24 h	21.08	[35]
	Bovine forage	*Saccharomyces cerevisiae* (3)	1261 μg/mL	48 h	20.00–55.00 (binding)	[69]
	Soil, rotten wood, olive, et al.	*Trichoderma* sp. (65)	50 ng/kg	7 d	17.80–100.00	[70]
	/	*Agrocybe cylindracea* GC-Ac2	100 ng/mL	37.9 h	96.00	[71]
	/	*Pleurotus ostreatus*	2500 ng/g	6 w	>80.00	[72]

Denote: “/” displays unknown information.

**Table 2 animals-15-03351-t002:** Microbial degrading enzymes of AFB1: their sources and functional conditions.

Enzyme		Producing Organism	Optimal Conditions	References
Laccase	BsCotA	*Bacillus subtilis*	pH 7.0, 70 °C, aerobic	[77,78]
	CotA-Laccase	*Bacillus licheniformis*	pH 2.5–4.5, 80–90 °C, aerobic	[79,80]
	B10 laccase	*Bacillus amyloliquefaciens* B10	pH 6.0–8.0, 40 °C, aerobic	[81]
	Lac-W	*Weizmannia coagulans* 36D1	pH 9.0, 30 °C, aerobic	[82]
	Lac 2	*Cerrena unicolor* 6884	pH 7.0, 45–55 °C, aerobic	[83]
	C30 laccase	*Saccharomyces cerevisiae*	pH 5.7, 30 °C, aerobic	[84]
	Ery4 laccase	*Pleurotus eryngii*	aerobic	[85]
	StMCO	*Streptomyces thermocarboxydus*	pH 4.0, aerobic	[86]
Peroxidase	IlMnP1,2,4,5,6, PcMnP1, CsMnP, and NfMnP	*Irpex lacteus*, *Phanerochaete chrysosporium*, *Ceriporiopsis subvermispora*, and *Nematoloma frowardii*	pH 3.0–4.5, 25–37 °C, they do not rely on O_2_ but utilize H_2_O_2_ as the electron acceptor.	[75,87]
	MnP	*Phanerochaete sordida* YK-624		[74]
	Il-MnP1, Il-DyP4	*Irpex lacteus* F17		[88]
	Rh_DypB	*Rhodococcus jostii*		[89]
	BsDyP	*Bacillus subtilis* SCK6		[90]
F_420_H_2_-dependent reductases	FDR-A and FDR-B	*Mycobacterium smegmatis* mc2155	Not rely on O_2_ but utilize NAD(P)H to provide the reduction equivalent.	[76]
Others	Aflatoxin Oxidase	*Armillariella tabescens*	aerobic	[91]
	*Bacillus* aflatoxin-degrading enzyme	*Bacillus shackletonii* L7	pH 8.0, 70 °C, aerobic	[92]
	Myxobacteria aflatoxin degradation enzyme	*Myxococcus fulvus* ANSM068	pH 6.0, 35 °C, aerobic	[93]
	N-acyl-homoserine lactonase	*Bacillus* sp.	pH 4.7, 37 °C, they do not rely on O_2_.	[24,94]
	Thermostable lipase	*Pseudomonas putida*	50–70 °C, they do not rely on O_2_.	[95]
	α/β hydrolase	*Bacillus halotolerans*	Do not rely on O_2_.	[34,96]
	Aldo/Keto reductase			
	Bacilysin bio-synthesis oxidoreductase	*Bacillus subtilis*	aerobic	[97]

**Table 3 animals-15-03351-t003:** Summary of the taxonomic composition of the microbiota associated with *Hermetia illucens*.

Substrates	Dominant Phyla	Dominant Genera	References
Food waste, calf forage, cooked rice	Actinobacteria, Bacteroidetes, Firmicutes, Fusobacteria, and Proteobacteria	*Citrobacter*, *Enterococcus*, *Klebsiella*, *Leminorella*, *Morganella*, and *Providencia*	[121]
Gainesville diet	Acidobacteria, Verrucomicrobia, Firmicutes, Actinobacteria, Proteobacteria, and Bacteroidetes	*Providencia*, *Bacteroides*, *Sphyngobacterium*, *Dysgonomonas*, and *Sanguibacter*	[122]
Gainesville diet	Firmicutes, Actinobacteria, Bacteroidetes, and Proteobacteria	*Bacillus*, *Cellulomonas*, *Empedobacter*, *Enterobacter*, *Gordonia*, *Kurthia*, *Microbacterium*, and *Micrococcus*	[123]
Chicken feed and vegetable waste	/	*Debaryomyces*, *Rhodotorula*, *Pichia*, *Geotrichum*, and *Trichosporon*	[116]
Chicken manure	Firmicutes, Proteobacteria, Bacteroidetes, and Actinobacteria	*Providencia*, *Enterococcus*, *Morganella*, and *Dysgonomonas*	[124]
Wheat bran containing tetracycline	Firmicutes, Proteobacteria, Bacteroidetes, Actinobacteria, and Fusobacteria	Bacteria: *Flavisolibacter*, *Proteus*, *Klebsiella*, *Actinomyces*, *Globicatella*, *Providencia*, *Enterococcus*, and *Ignatzschineria* Fungi: *Entyloma*, *Lysurus*, and *Trichophyton*	[125]
Mixture of vegetables and fish meal	Actinobacteria, Firmicutes, Bacteroidetes, and Proteobacteria	*Dysgonomonas*, *Providencia*, *Blautia*, *Shingobacterium*, *Morganella*, and *Bacillus*	[105]
Raw food waste	Firmicutes, Bacteroidetes, and Proteobacteria	*Bacillus*, *Lactobacillus*, *Dysgonomonas*, *Enterococcus*, and *Providencia*	[126]
Fruit/vegetable waste, manure, wheat bran, et al.	Proteobacteria and Firmicutes	*Morganella*, *Bacillus*, *Enterococcus*, *Providencia*, and *Lactobacillus*	[127]
Food waste and manure	Firmicutes, Bacteroidetes, and Proteobacteria	/	[128]
Soya meal supplemented with oxytetracycline	Firmicutes, Bacteroidetes, Actinobacteria, and Proteobacteria	*Enterococcus*, *Ignatzschineria*, *Providencia*, and *Morganella*	[129]
Swine/chicken manure	Firmicutes, Bacteroidetes, Actinobacteria, and Proteobacteria	*Enterococcus*, *Providencia*, *Morganella*, *Klebsiella*, *Ignatzschineria*, and *Clostridium*	[130]
Commercial chicken feed	Firmicutes, Bacteroidetes, Actinobacteria, and Proteobacteria	*Morganella*, *Klebsiella*, *Providencia*, *Enterobacter*, *Enterococcus*, *Bacillus*, *uncultured Lachnospiraceae*, *Actinomyces*, and *Dysgonomonas*	[131]
Mixture of wheat germ, alfalfa and corn flour	Proteobacteria, Firmicutes, Actinobacteria, and Bacteroidetes	*Providencia*, *Morganella*, *Klebsiella*, *Escherichia*, *Acinetobacter*, *Stenotrophomonas*, *Pseudomonas*, and *Enterococcus*	[132]
Chicken feed, grass, vegetables	Firmicutes, Bacteroidetes, Actinobacteria, and Proteobacteria	*Actinomyces*, *Dysgonomonas*, *Enterococcus*, and *unclassified Actinomycetales*	[133]
Mixture of mill waste, wheat bran, and alfa flour	Firmicutes, Bacteroidetes, Actinobacteria, and Proteobacteria	*Providencia*, *Klebsiella*, *Bacillus, Morganella, Alcaligenes*, *Bordetella*, and *Kerstersia*	[134]
Soy pulp and cafeteria waste	Bacteroidetes and Firmicutes	*Bacillus*, *Citrobacter*, *Dysgonomonas*, *Porphyromonas* and *Parabacteroides*	[135]
Wheat bran exposed to heavy metals (Cu and Cd)	Bacteroidetes, Actinobacteria, Proteobacteria, and Firmicutes	*Salana*, *Parabacteroidetes*, and *Campylobacter*	[136]
Hen diet, okara, maize distillers with soluble, brewer’s grains	Bacteroidetes, Actinobacteria, Proteobacteria, and Firmicutes	*Providencia*, *Morganella*, and *Klebsiella*	[137]
Chicken feed, cottonseed press cake	Proteobacteria and Firmicutes	Bacteria: *Enterobacteriaceae*, *Pseudomonas*, *Curtobacterium*, *Bacillus*, *Enterococcaceae*, and *Actinomycetaceae* Fungi: *Trichosporon*, *Cladosporium*, *Diutina*, *Aspergillus*, *Xeromyces*, and *Acaulium*	[118]
Chicken feed	Firmicutes, Bacteroidetes, Actinobacteria, and Proteobacteria	*Morganella*, *Enterococcus*, *Proteus*, *Providencia*, *Actinomyces*, *Lachnospiraceae*, *Enterobacteriaceae*, *Klebsiella*, *Escherichia-Shigella*, and *Citrobacter*	[138]
Food waste	Proteobacteria, Firmicutes, and Bacteroidetes	*Ignatzschineria*, *Providencia*, *Proteus*, *Klebsiella*, and *Vagococcus*	[139]
Chicken feed; fiber-rich ingredients	Proteobacteria, Firmicutes, Bacteroidetes, and Actinobacteria	*Enterococcus*, *Escherichia*, *Klebsiella*, *Providencia*, *Enterobacter*, and *Morganella*	[140]
Chicken feed; food waste; oil waste	Proteobacteria, Bacteroidetes, Firmicutes, and Actinobacteria	*Morganella*, *Providencia*, *Dysgonomonas*, *Lactobacillus*, and *Enterobacteriaceae*	[141]
Mixtures of coffee byproducts and microalgae	Firmicutes and Proteobacteria	*Morganella*, *Paenibacillus*, *Lysinibacillus*, *Brevundimonas*, *Enterococcus*, *Parococcus*, *Solibacillus*, and *Paracoccus*	[142]
Chicken manure	Firmicutes and Actinobacteria	Bacteria: *Unclassified_f_peptostreptococcaceae*, *Enterococcus*, and *Turicibacter* Fungi: *Penicillium*, *Aspergillus*, *Kernia,* and *Microascus*	[143]
Chicken manure; kitchen waste	Proteobacteria, Actinobacteria, and Firmicutes	*Providencia*, *Morganella*, *Brevibacterium*, *Staphylococcus*, and *Bordetella*	[144]
Chicken feed	Proteobacteria, Firmicutes, and Actinobacteria	*Providencia*, *Proteus*, *Morganella*, *Enterococcus*, *Bacillus*, and *Enterobacteriaceae*	[145]
Wheat bran, food waste and peanut shell	Firmicutes	*Bacillus*, *unclassified_f_Caloramatoraceae*, *Cerasibacillus*, and *Gracilibacillus*	[146]
Gainesville diet; starvation	Actinobacteria, Proteobacteria, Firmicutes, Euryarchaeota, and Bacteroidetes	*Actinomyces*, *Campylobacter*, *Microbacterium*, *Enterococcus*, and *Enterobacter*	[147]
Standard feed, brewer’s spent grain, plant-based sweetener, and vegetable waste	Firmicutes, Bacteroidetes, Actinobacteria, and Proteobacteria	*Morganella*, *Providencia*, *Lactobacillus*, *Enterococcus*, and *Proteus.*	[148]
Wheat bran and soybean powder, food waste	Proteobacteria, Firmicutes, Bacteroidetes, and Actinobacteria	*Enterococcus*, *Acinetobacter*, *Providencia*, *Enterobacter*, and *Myroides*	[149]
Corn flour and bran	Proteobacteria, Firmicutes, Bacteroidetes, and Actinomycetes	*Morganella*, *Sedimentibacter*, *Dysgonomonas*, *Enterococcus*, and *Providencia*	[150]
Chicken feed, mixture of vegetable coproducts, pig feed, and wheat bran	Firmicutes and Proteobacteria	*Lactiplantibacillus*, *Weissella*, *Enterococcus*, *Morganella*, *Providencia*, *Lactobacillus*, *Corynebacterium*, *Proteus*, *Oceanobacillus*, *Cerasibacillus*, *Enterobacter*, and *Bacillus*	[151]
Pig manure	Actinobacteria, Proteobacteria, and Bacteroidetes	*Enterococcus*, *Providencia*, *Dysgonomonas*, *Koukoulia*, *Pseudomonas*, *Sphingobacterium*, and *Clostridiaceae*	[152]
Palm kernel meal	Actinobacteria, Bacteroidetes, Firmicutes, and Proteobacteria	Bacteria: *Klebsiella*, *Enterococcus*, and *Sphingobacterium* Fungi: *Trichosporon*, *Candida*, *Lichtheimia*, *Fusarium*, *Pichia*, *Suhomyces*, *Diutina,* and *Kluyveromyces*	[117]
Household composts	/	*Nectriaceae*, *Meyerozyma*, *Kodamaeae*, *Gibberella*, *Diplodascaceae*, *Cyberlindnera*, and *Candida*	[153]
Gainesville diet	Actinobacteria, Firmicutes, Proteobacteria, and Bacteroidetes	*Acetobacter*, *Pseudomonas*, *Dysgonomonas*, *Acinetobacter*, *Providencia*, *Myroides*, *Alcaligenes*, and *Corynebacterium*	[154]
Corn Straw	Proteobacteria, Bacteroidetes, and Firmicutes	*Acinetobacter*, *Dysgonomonas*, and *unclassified Enterobacteriaceae*	[155]
Wheat bran, wheat middling	Proteobacteria, Firmicutes, and Bacteroidetes	Bacteria: *Dysgonomonas, Campylobacter*, *Enterococcus*, *Actinomyces*, *Pseudomonas*, *Klebsiella*, *Pediococcus*, *Lactobacillus*, *Bacillus*, *Orbus.*, and *Providencia* Fungi: *Issatchenkia*, *Candida*, *Aspergillus*, and *Wickerhamomyces*	[156]
Artificial diet with the addition of antibiotics	Proteobacteria, Firmicutes, and Actinobacteria	*Pseudomonas*, *Actinomyces*, *Morganella*, *Providencia*, and *Klebsiella*	[157]
Soya meal substrate containing oxytetracycline	/	*Enterococcus*, *Psychrobacter*, *Providencia*, *Myroides*, *Enterobacteriaceae*, and *Lactobacillales*	[158]
Mixtures of corn meal, wheat bran, and moisture content	Actinobacteria, Bacteroidetes, Firmicutes, and Proteobacteria	*Klebsiella*, *Clostridium*, *Acinetobacter*, *Pseudomonas*, *Providencia*, *Dysgonomonas*, *Morganella*, *Acetobacter*, *Enterococcus*, *Chryseobacterium*, and *Actinomyces*	[159]
Gainesville Diet	/	*Morganella*, *Dysgonomonas*, *Salmonella*, *Pseudochrobactrum*, and *Klebsiella* (12 °C); *Acinetobacter*, *Pseudochrobactrum*, *Enterococcus*, *Comamonas*, and *Leucobacter* (16 °C)	[160]

Denote: “/” represents unknown information.

## Data Availability

No new data were created or analyzed in this study. Data sharing is not applicable to this article.

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
