# Peer review of "Intestinal Microecological Mechanisms of Aflatoxin B1 Degradation by Black Soldier Fly Larvae (Hermetia illucens): A Review"

_animals, 2025, doi:10.3390/ani15223351_

Round 1
Reviewer 1 Report
Comments and Suggestions for Authors
Legislative framework is missing. around the world legislative framework is different and complicated. i.e. in EU it is still not allowed to use contaminated substrate for feeding insect intended for feed and food. A chapter dedicated would complete the MS, including limits and bottleneck related to the use of insect for matrix decontamination
The chapter 2 contains a lot of information however it is difficult for the readers to catch a clear take home message. A sum up of key concept per each paragraph would improve enormously the MS quality. Furtherly a real critical interpretation of literature results per each paragraph is missing. this aspect should be improved. moreoever in this session ametion to Toxins 2019, 11(11), 617; https://doi.org/10.3390/toxins11110617 is recommended since it reports the main approaches for Mycotoxin's decontaminations.
Typo L357: increased
3. Advantages of BSFL for the Degradation of AFB1 Waste Pollution: why Authors use waste pollution here
Author Response
Reviewer 1
Comments and Suggestions for Authors
Legislative framework is missing. around the world legislative framework is different and complicated. i.e. in EU it is still not allowed to use contaminated substrate for feeding insect intended for feed and food. A chapter dedicated would complete the MS, including limits and bottleneck related to the use of insect for matrix decontamination
Response: Thank you for this insightful suggestion. I have added a new part about legislative limitations and attached the following four references in lines 972-988 of revised manuscript, which provides the current regulations and substrate admission rules in the EU and US regarding the use of mycotoxin-contaminated substrates for insect rearing. The revised content is as follows:
The current application of BSFL for AFB1 degradation still faces notable legislative limitations, particularly the lack of targeted regulatory standards for its application in AFB1 contaminated feed or food-related matrices and the ambiguity in legal liability for product safety. For instance, European Union (EU) and the American Association of Feed Control Officers (AAFCO) legislation imposes strict regulations on the types, sources and application of substrates for insect farming, explicitly requiring the use of feed-grade raw materials from reliable sources [201-204]. Therefore, this regulatory requirement directly excludes mycotoxin-contaminated substrates from the scope of legally permissible substrates for BSFL cultivation, which are precisely the toxic substrates targeted by the “insect-mediated biotransformation of toxic substrates” technology [201,202]. Consequently, this technology lacks a feasible and compliant implementation pathway within the existing regulatory framework. Even though such waste can be degraded and recycled by insects, their large-scale application in insect farming remains prohibited due to legislative restrictions. In future, it is essential to advance the coordinated alignment between legislation and technical practices. Specifically, a feasible pathway for the large-scale application of BSFL-mediated AFB1 degradation can be established by defining safety standards and application boundaries for contaminated substrates.
Reference:
- Van Raamsdonk, L.W.D.; Van der Fels-Klerx, H.J.; De Jong, J. New feed ingredients: the insect opportunity. Food Addit. Contam. Part A 2017, 34, 1384-1397, doi:10.1080/19440049.2017.1306883.
- Pinotti, L.; Ottoboni, M. Substrate as insect feed for biomass production. J. Insects Food Feed 2021, 7, 585-596, doi:10.3920/JIFF2020.0110.
- Wang, Y.; Shelomi, M. Review of Black Soldier Fly (Hermetia illucens) as Animal Feed and Human Food. Foods 2017, 6, 91, doi:10.3390/foods6100091.
- Committee, E.S. Risk profile related to production and consumption of insects as food and feed. EFSA J. 2015, 13, 4257, doi:10.2903/J.EFSA.2015.4257.
The chapter 2 contains a lot of information however it is difficult for the readers to catch a clear take home message. A sum up of key concept per each paragraph would improve enormously the MS quality. Furtherly a real critical interpretation of literature results per each paragraph is missing. this aspect should be improved. moreoever in this session ametion to Toxins 2019, 11(11), 617; https://doi.org/10.3390/toxins11110617 is recommended since it reports the main approaches for Mycotoxin's decontaminations.
Response: Thank you for highlighting the lack of clear take home messages and critical interpretation in Chapter 2. I have inserted the concise key conclusion sentence at the beginning of each paragraph, and attached the following two references. The revised chapter cites the review presented in Toxins 2019, 11(11), 617 in line 129, 154, and 280. The revised content is as follows:
- Among them, physical methods enable rapid detoxification but are generally energy-intensive, while chemical methods can disrupt toxin molecular structures yet are accompanied by nutrient degradation and secondary toxic residues [13]. Both two approaches share inherent limitations and cannot simultaneously achieve the triple objectives of high efficiency, environmental friendliness, and safety.(lines 122-131 in the revised manuscript)
- Biological detoxification relies on the adsorption of AFB1 by microbial cells walls and the catalytic degradation of enzymes to reduce the toxicity of AFB1, and it offers distinct advantages such as operating under mild conditions, high specificity, and the absence of toxic residues [13]. However, their degradation efficiency is constrained by the complexity of the treated matrix, which has posed a bottleneck in their large-scale expansion. (lines 151-155 in the revised manuscript)
- However, most studies on microbial degradation of AFB1, whether in single or consortia, only employ pure culture-buffer systems; in contrast, practical contaminated matrices (e.g., feed, food, and waste) generally exhibit complex compositions with high lipids and high protein contents, exhibiting significant differences from the former in terms of system complexity and composition [27]. The practical application of these conclusions in real scenarios is therefore limited. (lines 194-199 in the revised manuscript)
- Enzymatic degradation is also one of the common methods for AFB1 detoxification. (line 245 in the revised manuscript)
Reference:
- Čolović, R.; Puvača, N.; Cheli, F.; Avantaggiato, G.; Greco, D.; Đuragić, O.; Kos, J.; Pinotti, L. Decontamination of Mycotoxin-Contaminated Feedstuffs and Compound Feed. Toxins 2019, 11, 617, doi:10.3390/toxins11110617.
- Verheecke, C.; Liboz, T.; Mathieu, F. Microbial degradation of aflatoxin B1: Current status and future advances. Int. J. Food Microbiol. 2016, 237, 1-9, doi:10.1016/j.ijfoodmicro.2016.07.028.
Advantages of BSFL for the Degradation of AFB1 Waste Pollution: why Authors use waste pollution here
Response: Thank you for pointing this out. I have revised the manuscript by replacing the collocation “waste pollution” with the more academic terms “AFB1 Contaminated Waste” throughout (lines 330-331 in the revised manuscript).

Reviewer 2 Report
Comments and Suggestions for Authors
The review «Intestinal Microecological Mechanisms of Aflatoxin B1 Degradation by Black Soldier Fly Larvae (Hermetia illucens): A Review» submitted for consideration is written on a topical topic. Indeed, there is currently scant and scattered information on the use of BSFL as a degradation tool for substances with high carcinogenicity and toxicity. The review is written in clear language, full of content and corresponds to the topic. The current review is relevant and of interest to the scientific community. The authors' materials expand the understanding of the use of BSFL as a biological tool for solving the issue of waste disposal with a high toxic load. The article is drawn up in accordance with the requirements of the journal, includes all the necessary sections. Sequential presentation of the material: characterization of Aflatoxin, description of existing and promising detoxification pathways, careful description and analysis of microbiological and enzymatic pathways for the degradation of aflatoxin into low-toxic metabolites, Synergistic Degradation of AFB1 by Endogenous Enzymes and Intestinal Microbiota of BSFL. Potential degradation pathways of AFB1 by BSFL intestinal microorganisms. Most cited sources are recently published and relevant. It is necessary to reflect how the authors selected the literature, what databases were used, what methodological and digital means were used in the work. The statements and conclusions made are supported by quotes, consistent. Drawings/tables/images/diagrams are appropriate, concise, correctly represent data. They are easy to interpret and understand.
Thus, the publication fully discloses the degradation potential of AFB1 by BSFL. BSFLs can maintain normal growth and development in mycotoxin-contaminated substrates and effectively degrade AFB1 without residues in their bodies, thereby avoiding the risk of secondary toxin accumulation in food chains. These BSFL characteristics provide practical and safe technical support for their use in detoxifying AFB1 contaminants in complex organic waste matrices. In general, the reading of the article causes a positive impression, contains a description of AFB1 by BSFL degradation models, which should be further taken into account in the application aspect when producing BSFL.
An important addition to the material would be to refine the management of the BSFL microbiota to increase the biodegradation potential of toxic substances. It is important to consider the residual amounts of toxin in the dried biomass and in the end products from BSFL. Animal feed based on BSFL should contain a minimum amount of Aflatoxin, it would be relevant to provide this data in the article. But this proposal is of a recommendatory nature, the material presented by the authors is quite enough and it represents a high contribution. The materials make it possible to further deepen research in this area, to use insects as a tool for biodegrading toxic substances in waste processing cycles.
Author Response
Reviewer 2
Comments and Suggestions for Authors
The review «Intestinal Microecological Mechanisms of Aflatoxin B1 Degradation by Black Soldier Fly Larvae (Hermetia illucens): A Review» submitted for consideration is written on a topical topic. Indeed, there is currently scant and scattered information on the use of BSFL as a degradation tool for substances with high carcinogenicity and toxicity. The review is written in clear language, full of content and corresponds to the topic. The current review is relevant and of interest to the scientific community. The authors' materials expand the understanding of the use of BSFL as a biological tool for solving the issue of waste disposal with a high toxic load. The article is drawn up in accordance with the requirements of the journal, includes all the necessary sections. Sequential presentation of the material: characterization of Aflatoxin, description of existing and promising detoxification pathways, careful description and analysis of microbiological and enzymatic pathways for the degradation of aflatoxin into low-toxic metabolites, Synergistic Degradation of AFB1 by Endogenous Enzymes and Intestinal Microbiota of BSFL. Potential degradation pathways of AFB1 by BSFL intestinal microorganisms. Most cited sources are recently published and relevant. It is necessary to reflect how the authors selected the literature, what databases were used, what methodological and digital means were used in the work.
Response: Thank you for pointing this out. Database includes National Center for Biotechnology Information (NCBI, https://www.ncbi.nlm.nih.gov/gdv/), PubMed (MEDLINE) database (https://pubmed.ncbi.nlm.nih.gov/), Web of Science Core Collection (https://www.webofscience.com), and KEGG pathway Database (https://www.genome.jp /kegg/). I have added these databases in lines 105-111 of revised manuscript. The revised content is as follows:
The present review summarizes the studies on the detoxification methods for AFB1 degradation, with a particular focus on microbial degradation processes, and the research progress regarding the sequencing of core intestinal microorganisms in BSFL reported in the databases, such as National Center for Biotechnology Information (NCBI, https://www.ncbi.nlm.nih.gov/gdv/), PubMed (MEDLINE) database (https://pubmed.ncbi. nlm.nih.gov/), Web of Science Core Collection (https://www.webofscience.com), and KEGG pathway database (https://www.genome.jp/kegg/).
The statements and conclusions made are supported by quotes, consistent. Drawings/tables/images/diagrams are appropriate, concise, correctly represent data. They are easy to interpret and understand. Thus, the publication fully discloses the degradation potential of AFB1 by BSFL. BSFLs can maintain normal growth and development in mycotoxin-contaminated substrates and effectively degrade AFB1 without residues in their bodies, thereby avoiding the risk of secondary toxin accumulation in food chains. These BSFL characteristics provide practical and safe technical support for their use in detoxifying AFB1 contaminants in complex organic waste matrices. In general, the reading of the article causes a positive impression, contains a description of AFB1 by BSFL degradation models, which should be further taken into account in the application aspect when producing BSFL.
Response: Thank you for this suggestion. To further address the application aspect, I have added a discuss about the necessity of validating the degradation model under commercial rearing conditions in lines 972-977 of the revised manuscript. The revised content is as follows:
Moreover, it is necessary to verify the stability of the BSFL-mediated degradation model under large-scale rearing conditions. Key indicators such as the degradation rate, product residue, and microbial safety limits can be clearly defined to establish a large-scale, standardized biodegradation process and a complete risk prevention and control system, thereby providing a viable pathway for the application of BSFL in organic waste treatment.
An important addition to the material would be to refine the management of the BSFL microbiota to increase the biodegradation potential of toxic substances.
Response: Thank you for this suggestion. I have added sentences to the revised manuscript that propose future work should modulate the BSFL microbiota to further enhance AFB1 biodegradation (lines 966-970). The revised content is as follows:
Meanwhile, the detoxification capacity of intact BSFL can be further enhanced by modulating their intestinal microbiota. For instance, the inoculation of AFB1-degrading strains isolated from the BSFL gut into the rearing substrate, or the enrichment of indigenous degrading bacteria through dietary manipulation, serves as effective strategies.
It is important to consider the residual amounts of toxin in the dried biomass and in the end products from BSFL. Animal feed based on BSFL should contain a minimum amount of Aflatoxin, it would be relevant to provide this data in the article. But this proposal is of a recommendatory nature, the material presented by the authors is quite enough and it represents a high contribution. The materials make it possible to further deepen research in this area, to use insects as a tool for biodegrading toxic substances in waste processing cycles.
Response: Thank you for this suggestion. I have added a discusse about the residual toxin levels in dried biomass in the revised manuscript. After BSFL treated the substrate contaminated with AFB1, the AFB1 concentration in the dried or freeze-dried larvae was lower than the detection limit (i.e., 0.10 μg/kg), which was far below the limit standard concentration (20 µg/kg), thus demonstrating product safety (lines 371-374). We also recommend establishing a complete risk prevention and control system during scale-up to ensure the safety and feasibility of BSFL-based detoxification (lines 966-971). The revised content is as follows:
BSFL effectively degraded 83%-95.1% of AFB1 while maintaining a high survival rate in a feed matrix supplemented with 0.415 mg/kg AFB1, and the AFB1 concentrations in the freeze-dried larvae fell below the analytical detection threshold (<0.10 µg/kg) [101]. (lines 371-374)
Moreover, it is necessary to verify the stability of the BSFL-mediated degradation model under large-scale rearing conditions. Key indicators such as the degradation rate, product residue, and microbial safety limits can be clearly defined to establish a large-scale, standardized biodegradation process and a complete risk prevention and control system, thereby providing a viable pathway for the application of BSFL in organic waste treatment. (lines 972-977)

Reviewer 3 Report
Comments and Suggestions for Authors
Abstract
I do not completely to understand the final idea of the abstract: “Given the low cost, high efficiency, safety, and sustainability of using the BSFL intestinal microbiota for AFB1 degradation, this study provides a scientific reference for managing AFB1 pollution to overcome food security issues.” How it is possible to manage the AFB pollution by using the intestinal microbiota? By its separate cultivation? By cultivation of the animals with such intestinal microbiota? I recommend to clarify a little bit the idea.
Main text
The Review somewhat violates the logic of motivation in favor of large-scale detoxification of AFB1 through intensive breeding:
1. The authors discuss various microorganisms in the role of AFB1b destructors, including many aerobic bacteria. Among the enzymes, the authors point to oxidases, in particular, laccases, which require aerobic conditions for biocatalysis and presence of filamentous fungal producers, but not yeast cells (they are absent in Figure 2). In addition, there are anaerobic conditions in intestine, so the oxidation processes are not enough active. The individual fungal enzymes discussed in the article mainly work at pH 4.5-5.0, and what is value of pH is in the intestine of BSFL?
2. At the end of Chapter 2, the authors conclude about the benefits of using BSFL, since these insects “survive and degrade AFB1 under more diverse, complex, and even harsh environmental conditions and achieve large-scale application through intensive breeding. This approach thus provides an efficient, economical, and sustainable solution to AFB1 pollution.” Why do the authors consider individual enzymes and individual strains of microorganisms, rather than their consortia, as an alternative to using BSFL gut? If someone does not use an animal with microorganisms in the intestine to destroy BSFL, but separate consortia of microorganisms, the number of which is very easy to increase outside the BSFL intestine, then the benefit may not be same what the authors write about. This option should also be discussed in the Review.
Lines 439-440: Please, add the references to the phrase: “In the intestinal tract of BSFL, microorganisms, such as Bacillus, Aspergillus, Pseudomonas, Escherichia coli, Stenotrophomonas, and Enterococcus, can degrade AFB1.” Please, put in correlation another phrase from the same text (Lines 282-283): “In existing studies, the fungal species in the intestinal tract of BSFL mainly belong to the phylum Ascomycota, including the genera Pichia, Candida, Diutina, Cyberlindnera, Aspergillus, Geotrichum, and Trichosporon [106-110].” Most of these fungi absent in Figure 2. Where are the representatives of Aspergillus genus here?
Table 1: There are some data which are performed in the form of following units “ppm” or “ppb”, whereas the most rest part of information is presented by using traditional units like “μg/mL”, “mg/L”, “ng/mL” or “μg/kg”. So, I recommend to use single approach to the data presentation. Additionally, he “sp.” in all names of microorganisms should not be italicized.
There are a lot of places in the text, where the names of authors of articles and numbers of corresponding references are mentioned at the same time: Arun et al. [21], Bovo et al. [20], Gonzalez et al. [23], Petchkongkaew et al. [24], Wang et al. [25], Suo et al. [44], Bosch et al. [96], Purschke et al. [102], Meijer et al. [103], Camenzuli et al. [104]. According to the recommendations of MDPI only numbers of the references should present in the text of the article. Please, remove all names from the text.
References
Totally:
- Please give the commonly accepted abbreviations instead of complete titles of the Journals in the references.
- Please, check the correctness of the writing of latin names of the living organisms (including the List with all references): the names should be italicized and when you give the genusandspecies of the microorganism the species should start from little letter. Please, see:
- Line635: Bacillus
- Lines 640: Bacillus licheniformis
- Line 643: Bacillus subtilis
- Lines 645: Stenotrophomonas Maltophilia; Line 648: Bacillus aryabhatta
- Line 658: Bacillus megaterium
- Line 664: Aspergillus flavus
- Line 669: Pseudomonas
- Line 670: Pseudomonas putida
- Line 672: Pseudomonas knackmussii
- Lines 678, 681: Stenotrophomonas
- Line 684: Stenotrophomonas acidaminiphila
- Line 688: Rhodococcus erythropolis
- Line 691: Rhodococcus
- Line 693: Rhodococcus pyridinivorans
- Line 695: Rhodococcus turbidus, etc.
Please check all 175 links, almost every one of them has a latin name!
Lines 466-470: there is same situation with latin names of microorganisms. It should be corrected.
Reference 9: I recommend to replace it. Bibliography is incomplete in it.
The phrase “Supplementary Materials: No more supplement.” can be removed from the article.
Author Response
Reviewer 3
Comments and Suggestions for Authors
Abstract
I do not completely to understand the final idea of the abstract: “Given the low cost, high efficiency, safety, and sustainability of using the BSFL intestinal microbiota for AFB1 degradation, this study provides a scientific reference for managing AFB1 pollution to overcome food security issues.” How it is possible to manage the AFB pollution by using the intestinal microbiota? By its separate cultivation? By cultivation of the animals with such intestinal microbiota? I recommend to clarify a little bit the idea.
Response: Thank you for this helpful comment. I have now clarified in the abstract section of the revised manuscript (lines 42-46) that the BSFL itself functions as a living microbial reactor in which the functional flora and the larval host detoxification enzyme system synergistically degrade AFB1 in the rearing substrate, without the need for separate microbial cultivation. This synergistic mechanism offers a low-cost, efficient, and sustainable strategy for controlling AFB1 contamination in organic waste streams. To remove the ambiguity, replace the original closing sentence with the below description:
Given the low cost, high efficiency, safety, and sustainability of using the BSFL as living microbial reactors in which the core gut microbiota and the larval host detoxifying enzyme system synergistically degrade AFB1, this study provides a scientific reference for managing AFB1 pollution to overcome food security issues.
Main text
The Review somewhat violates the logic of motivation in favor of large-scale detoxification of AFB1 through intensive breeding:
- The authors discuss various microorganisms in the role of AFB1b destructors, including many aerobic bacteria. Among the enzymes, the authors point to oxidases, in particular, laccases, which require aerobic conditions for biocatalysis and presence of filamentous fungal producers, but not yeast cells (they are absent in Figure 2). In addition, there are anaerobic conditions in intestine, so the oxidation processes are not enough active.
Response: Thank you for pointing out this critical issue. Considering the anoxic environment in BSFL lumen (≤ 2.5% O2), I have re-evaluated the candidate microorganisms and enzymes in the revised manuscript. Laccases and laccase-like enzymes such as BsCotA, CotA-Laccase, B10 laccase, Lac-W, Lac2, C30 laccase, Ery4 laccase, and StMCO, are aerobic enzymes that are essentially inactive in the central anoxic lumen. Instead, I emphasise oxygen-independent pathways: α/β-hydrolase and AHL-lactonase (hydrolytic), DyP peroxidase (uses trace H2O2), AKR (NADPH-dependent reduction), and yeast-cell-wall adsorption, all of which retain high AFB1-degrading capacity under anoxic conditions. The issue of whether enzymes require oxygen has been supplemented in Table 1. The metabolic pathways in Figure 2, 3 and the Graphical abstract have also been modified accordingly based on whether they can function in the anaerobic intestinal environment. The revised content is as follows:
- CotA and BacC are aerobic enzymes, and their activity is low in the central anoxic lumen (≤5% O2). On the contrary, ABH and AHL-lactonase (hydrolytic), DyP peroxidase (uses trace H2O2), and AKR (NADPH-dependent reduction), all of which retain high AFB1-degrading capacity under anoxic conditions [24,34,87,94-96]. (lines 554-558)
- In addition, strains of Myroides, Escherichia, Staphylococcus, Lysinibacillus, Enterobacter, Klebsiella, Aspergillus, Microbacterium, and Enterococcushave been demonstrated to harbor AFB1-detoxifying capacities (Table 1), and these genera are frequently detected in intestinal microbiota analyses of BSFL (Table 3). Considering the anoxic environment of the BSFL intestinal tract, facultative anaerobic bacteria including Bacillus, Pseudomonas, Escherichia, Stenotrophomonas, Enterococcus, Staphylococcus, Enterobacter, and Klebsiella can survive and exert metabolic activities in the anoxic intestinal segments of BSFL [170-177]. Therefore, they may directly participate in AFB1 degradation mediated by the BSFL intestinal microbiota. On the contrary, Aspergillus, Myroides, Lysinibacillus, and Microbacterium are aerobes [178-181]. Their mycelial growth or respiratory chains require continuous oxygen supply, rendering them unable to perform active oxidative metabolism of AFB1 in the anoxic central segments of the intestinal lumen. (lines 567-578)
- In addition to direct degradation, lactic acid bacteria (e.g., Lactobacillus, Weissella, and Pediococcus) and yeast (e.g., Pichia, Candida,Geotrichum, and Trichosporon) adsorb AFB1 via components such as peptidoglycan, teichoic acid, and β-glucan in their cell walls [20,21,56,156,182-187] (Figure 2D). Lactobacillus, Weissella, Pediococcus, Pichia, Candida, Geotrichum, and Trichosporon are all facultative anaerobes that can survive and exert adsorptive activity in the anoxic intestinal tract of BSFL, thereby reducing the bioavailability of AFB1 in the intestinal tract and affording protection against injury to the host [87,188-192]. (lines 592-604)
- The current research on the biodegradation of AFB1 mainly focuses on redox reactions as the dominant mechanism. The unsaturated sites on the furan ring, lactone ring, and cyclopentenone ring of AFB1 undergo oxidation, reduction, or hydroxylation reactions, which block the binding ability of AFB1 to DNA, thereby reducing its carcinogenic risk and forming metabolites with low stability. Among the main pathways for AFB1 biodegradation is an oxygenation reaction to yield highly toxic AFBO, which is subsequently converted into the less toxic AFB1-dihydrodiol [73,74]. This step can be catalyzed by the endogenous detoxification enzyme system of BSFL. Bacillus subtilis, Bacillus licheniformis, and Bacillus amyloliquefaciensconvert AFB1 into its hydroxyl derivative Aflatoxin Q1 (AFQ1) via the secretion of laccase, whose toxicity and mutagenicity are 1/18 and 1% of those of AFB1, respectively [77,81,194-196]. Aspergillus niger not only reduces the ketone carbonyl group on the cyclopentenone ring of AFB1 to form the AFL, but also mediates the reductive elimination of AFB1 after hydrolysis and ring opening of its lactone ring [197-198]. The BacC enzyme secreted by Bacillus subtilis can reduce the α,β-unsaturated ester moiety of AFB1 [97]. After these reactions, the key toxic sites of AFB1 are destroyed, resulting in a significant reduction in its carcinogenicity and mutagenicity. However, in the microaerophilic or anaerobic intestinal environment of BSFL, oxidative reactions exhibit low activity, and the degradation of AFB1 primarily relies on non-oxidation catalyzed reactions mediated by the intestinal anaerobic or facultative anaerobic microbial community. (lines 615-634)
- Apart from these minor redox reactions, hydrolysis and cracking reactions play a dominant role. After hydrolysis and cracking reactions, further reactions such as demethylation, demethoxylation, or decarbonylation gradually cleave it into various low-toxicity or nontoxic small-molecule products, thereby reducing the residue and toxicity of AFB1 in the environment [46,199].(lines 643-647)
- The revised Figure 2 in the updated manuscript is shown below.

- The revised Figure 3 in the updated manuscript is shown below.

- The revised Graphical abstract in the updated manuscript is shown below.

Reference:
- Dragičević, T.; Hren, M.; Gmajnić, M.; Pelko, S.; Kungulovski, D.; Kungulovski, I.; Čvek, D.; Frece, J.; Markov, K.; Delaš, F. Biodegradation of Olive Mill Wastewater by Trichosporon Cutaneumand Geotrichum Candidum. Arh. Hig. Rada. Toksikol. 2010, 61, 399-405, doi:10.2478/10004-1254-61-2010-2079.
- Wang, L.; Weng, L.; Dong, Y.; Zhang, L. Specificity and Enzyme Kinetics of the Quorum-quenching N-Acyl Homoserine Lactone Lactonase (AHL-lactonase). J. Biol. Chem. 2004, 279, 13645-13651, doi:10.1074/jbc.M311194200.
- Bui, S.; Gil-Guerrero, S.; Van der Linden, P.; Carpentier, P.; Ceccarelli, M.; Jambrina, P.G.; Steiner, R.A. Evolutionary adaptation from hydrolytic to oxygenolytic catalysis at the α/β-hydrolase fold. Chem. Sci. 2023, 14, 10547-10560, doi:10.1039/d3sc03044j.
- Beranová, J.; Mansilla, M.C.; Mendoza, D.d.; Elhottová, D.; Konopásek, I. Differences in Cold Adaptation of Bacillus subtilisunder Anaerobic and Aerobic Conditions. J. Bacteriol. 2010, 192, 4164-4171, doi:doi:10.1128/jb.00384-10.
- Wu, M.; Guina, T.; Brittnacher, M.; Nguyen, H.; Eng, J.; Miller, S.I. The Pseudomonas aeruginosaProteome during Anaerobic Growth. J. Bacteriol. 2005, 187, 8185-8190, doi:10.1128/jb.187.23.8185-8190.2005.
- Tseng, C.P.; Albrecht, J.; Gunsalus, R.P. Effect of microaerophilic cell growth conditions on expression of the aerobic (cyoABCDEand cydAB) and anaerobic (narGHJI, frdABCD, and dmsABC) respiratory pathway genes in Escherichia coli. J. Bacteriol. 1996, 178, 1094-1098, doi:10.1128/jb.178.4.1094-1098.1996.
- Gupta, S.; Goel, S.S.; Siebner, H.; Ronen, Z.; Ramanathan, G. Transformation of 2, 4, 6-trinitrotoluene byStenotrophomonasstrain SG1 under aerobic and anaerobic conditions. Chemosphere 2023, 311, 137085, doi:10.1016/j.chemosphere.2022.137085.
- Al-Fatlawi, A.H.; Raheem, S.A. Inactivation of Enterococcus faecalisin drinking water using silver nanoparticles embedded paper. Indian J. Forensic Med. Toxicol. 2020, 14, 1117-1121, doi:10.37506/v14/i1/2020/ijfmt/193057.
- Song, J.; Sun, D.; Zhao, L.; Jiang, H.; Zhu, C. High Power Generation by a Strain of Facultative Anaerobe in Double-Chamber Microbial Fuel Cell. Adv. Mater. Res. 2012, 347, 2616-2621, doi:10.4028/www.scientific.net/AMR.347-353.2616.
- Singh, S.; Singh, A.K.; Singh, S.K.; Yadav, V.B.; Kumar, A.; Nath, G. Current update on the antibiotic resistance profile and the emergence of colistin resistance in Enterobacter isolates from hospital-acquired infections.Microbe 2025, 8, 100432, doi:10.1016/j.microb.2025.100432.
- Kong, D.; Park, J.; Lee, C.; Khandelwal, H.; Kim, M.; Sakuntala, M.; Kim, T.; Jeon, B.; Kim, J.; Kim, C. Reprint of “A newly isolated Klebsiella variicolaJYP01 strain with iron-interaction capability for energy-efficient production of 1,3-propanediol”. J. Taiwan Inst. Chem. Eng. 2025, 177, 106443, doi:10.1016/j.jtice.2025.106443.
- Tarrand, J.J.; Han, X.; Kontoyiannis, D.P.; May, G.S. Aspergillus Hyphae in Infected Tissue: Evidence of Physiologic Adaptation and Effect on Culture Recovery. J. Clin. Microbiol. 2005, 43, 382-386, doi:10.1128/jcm.43.1.382-386.2005.
- Yang, S.; Liu, Q.; Shen, Z.; Wang, H.; He, L. Molecular Epidemiology of Myroides odoratimimusin Nosocomial Catheter-Related Infection at a General Hospital in China. Infect. Drug Resist. 2020, 13, 1981-1993, doi:10.2147/IDR.S251626.
- Liu, H.; Song, Y.; Chen, F.; Zheng, S.; Wang, G. Lysinibacillus manganicussp. nov., isolated from manganese mining soil. Int. J. Syst. Evol. Microbiol. 2013, 63, 3568-3573, doi:10.1099/ijs.0.050492-0.
- Egorova, D.O.; Demakov, V.A.; Plotnikova, E.G. Bioaugmentation of a polychlorobiphenyl contaminated soil with two aerobic bacterial strains. J. Hazard. Mater. 2013, 261, 378-386, doi:10.1016/j.jhazmat.2013.07.067.
- Repečkienė, J.; Levinskaitė, L.; Paškevičius, A.; Raudonienė, V. Toxin-producing fungi on feed grains and application of yeasts for their detoxification. Pol. J. Vet. Sci. 2013, 16, 391-393, doi:10.2478/pjvs-2013-0054.
- Marlida, Y.; Harnentis; Anggraini, L.; Ardani, L.; Huda, N. Yeast Probiotic Isolated from Fish Fermented (Budu) with Promising AFB1 Biodetoxify. Int. J. Vet. Sci. 2025, 14, 310-315, doi:10.47278/journal.ijvs/2024.253.
- Bzducha Wróbel, A.; Bryła, M.; Gientka, I.; Błażejak, S.; Janowicz, M. Candida utilis ATCC 9950 Cell Walls and β(1,3)/(1,6)-Glucan Preparations Produced Using Agro-Waste as a Mycotoxins Trap. Toxins 2019, 11, 192, doi:10.3390/toxins11040192.
- Sidari, R.; Tofalo, R. Dual Role of Yeasts and Filamentous Fungi in Fermented Sausages. Foods 2024, 13, 2547, doi:10.3390/foods13162547.
- Rodríguez-Rivera, V.; Estrada-García, J.; Sales-Pérez, R.E.; Hernández-Martínez, J.M.; Méndez-Contreras, J.M. Valorization of Agro-industrial Waste to Produce a Probiotic-bio-stimulant Through the Anaerobic Co-fermentation Process With Lactobacillus casei: a Circular Economy Approach in Vulnerable Communities of Mexico. Water, Air, Soil Pollut. 2025, 236, 925, doi:10.1007/s11270-025-08587-7.
- Diez, A.M.; Björkroth, J.; Jaime, I.; Rovira, J. Microbial, sensory and volatile changes during the anaerobic cold storage of morcilla de Burgos previously inoculated with Weissella viridescensand Leuconostoc mesenteroides. Int. J. Food Microbiol. 2009, 131, 168-177, doi:10.1016/j.ijfoodmicro.2009.02.019.
- Sun, J.; Zhang, Y.; Zhao, Y.; Wang, Z.; Miao, X.; Huo, W.; Chen, L.; Liu, Q.; Wang, C.; Guo, G. Enhancing Alfalfa Hemicellulose Degradation by Anaerobic Bioprocessing with Engineered Xylanase-Secreting Pediococcus pentosaceus. J. Agric. Food. Chem. 2025, 73, 22563-22576, doi:10.1021/acs.jafc.5c05169.
- Sumi, A.; Morimura, S.; Shigematsu, T.; Takenouchi, H.; Kida, K. Anaerobic Digestion of Wastewater Including High Concentration of Yeast, Pichia pastoris. Japanese J. Wat. Treat. Biol. 2005, 41, 213-218, doi:10.2521/jswtb.41.213.
- Wenda, J.M.; Drzewicka, K.; Mulica, P.; Tetaud, E.; di Rago, J.P.; Golik, P.; Łabędzka-Dmoch, K. Candida albicansPPR proteins are required for the expression of respiratory Complex I subunits. Genetics 2024, 228, iyae124, doi:10.1093/genetics/iyae124.
The individual fungal enzymes discussed in the article mainly work at pH 4.5-5.0, and what is value of pH is in the intestine of BSFL?
Response: Thank you for pointing this out. The original manuscript already points out that the luminal pH of the BSFL midgut show a gradient from weakly acidic (pH=5.9) to strongly acidic (pH=2.1) to alkaline (pH=8.3). The pH values in other areas are alkaline, ranging from 6 to 8. Therefore, the fungal enzymes that are most active at pH 4.5-5.0 can function in the anterior midgut, but their catalytic efficiency drops in the strongly acidic midgut centre (pH=2.1) and in the alkaline posterior segments (pH 6-8.3). This compartmental match indicates that acidic-tolerant enzymes contribute mainly to the initial attack on AFB1, whereas neutral and alkaline enzymatic mechanisms dominate in downstream gut sections.
References:
- 1. Bonelli, M.; Bruno, D.; Caccia, S.; Sgambetterra, G.; Cappellozza, S.; Jucker, C.; Tettamanti, G.; Casartelli, M. Structural and Functional Characterization of Hermetia illucensLarval Midgut. Front. Physiol. 2019, 10, 204, doi:10.3389/fphys.2019.00204.
- Callegari, M. The gut microbiome associated to honeybees and waste-reducing insects. 2017.
- At the end of Chapter 2, the authors conclude about the benefits of using BSFL, since these insects “survive and degrade AFB1 under more diverse, complex, and even harsh environmental conditions and achieve large-scale application through intensive breeding. This approach thus provides an efficient, economical, and sustainable solution to AFB1 pollution.” Why do the authors consider individual enzymes and individual strains of microorganisms, rather than their consortia, as an alternative to using BSFL gut?
Response: Thank you for your correction. After verification, it was found that the original manuscript does have omissions in its expression. The following paragraph has now been added at the end of Chapter 2 (lines 277-297):
Current research on the biodegradation of AFB1 mostly focuses on detoxifying microorganisms or enzymes in vitro. Although laboratory studies have extensively explored aflatoxin degradation, a mature biological solution for full-scale commercial application remains absent [13]. Various nutrients and inhibitors present in actual contaminated substrates can readily alter the pH, ionic strength, and optimal temperature range of microorganisms or enzymes whether in single or consortia, thereby reducing degradation efficiency [31]. Compared with the use of microbiota or enzymes to degrade AFB1, insects have stronger environmental stress resistance and broader substrate adaptability. They can survive and degrade AFB1 under more diverse, complex, and even harsh environmental conditions and achieve large-scale application through intensive breeding. This approach thus provides an efficient, economical, and sustainable solution to AFB1 pollution.
References:
- Čolović, R.; Puvača, N.; Cheli, F.; Avantaggiato, G.; Greco, D.; Đuragić, O.; Kos, J.; Pinotti, L. Decontamination of Mycotoxin-Contaminated Feedstuffs and Compound Feed. Toxins 2019, 11, 617, doi:10.3390/toxins11110617.
If someone does not use an animal with microorganisms in the intestine to destroy BSFL, but separate consortia of microorganisms, the number of which is very easy to increase outside the BSFL intestine, then the benefit may not be same what the authors write about. This option should also be discussed in the Review.
Response: Thank you for raising this question. I have added a discuss about the advantages of the synergistic degradation system of BSFL and intestinal microorganisms over the consortium in vitro (lines 314-329). I fully agree that the independent scaled-up cultivation of BSFL-derived degrading microbiota is indeed a feasible and scalable approach. However, the synergistic system composed of BSFL and their intestinal microbiota enables efficient and sustaining AFB1 degradation through the integrated metabolism of endogenous enzymes and intestinal bacteria, the regulatory role of immune function in maintaining intestinal microbial composition, and the in situ adaptive remodeling of microbiota, thus demonstrating significant superiority over the traditional large-scale in vitro complex microbial fermentation model. The revised content is as follows:
Therefore, BSFL can not only be employed as living microbial reactors for AFB1 degradation, but their gut-derived microbiota can also be cultivated in vitro as a microbial detoxification agent. However, compared with the traditional large-scale in vitro complex microbial fermentation model, the synergistic system composed of BSFL and their intestinal microbiota enables efficient and sustaining AFB1 degradation through the integrated metabolism of endogenous enzymes and intestinal bacteria, the regulatory role of immune function in preserving intestinal microbial composition, and the in situ adaptive remodeling of microbiota, thus conferring significant superiority [104-107]. On the contrary, in vitro cultured complex microbial consortia require specific induction, subculture, or genetic engineering to achieve such an effect, and are susceptible to contamination, leading to substantial and often irreversible declines in degradation efficiency. Moreover, the intestinal lumen of BSFL exhibits an alkaline-weakly acidic-strongly acidic-alkaline pH gradient [108]. Different types of degrading microorganisms and enzymes can continuously degrade AFB1 under this spatial distribution, while it is challenging to replicate such a pH gradient in a single fermentation vessel for in vitro cultured complex microbial consortia.
Lines 439-440: Please, add the references to the phrase: “In the intestinal tract of BSFL, microorganisms, such as Bacillus, Aspergillus, Pseudomonas, Escherichia coli, Stenotrophomonas, and Enterococcus, can degrade AFB1.”
Response: Thank you for pointing this out. I have now inserted the supporting references [41,46,57,90,118,132,197] after the sentence in line 637 of the revised manuscript.
References:
- Samuel, M.S.; Sivaramakrishna, A.; Mehta, A. Degradation and detoxification of aflatoxin B1 by Pseudomonas putida. Int. Biodeterior. Biodegrad. 2014, 86, 202-209, doi:10.1016/j.ibiod.2013.08.026.
- Suo, J.; Liang, T.; Zhang, H.; Liu, K.; Li, X.; Xu, K.; Guo, J.; Luo, Q.; Yang, S. Characteristics of aflatoxin B1 degradation by Stenotrophomonas acidaminiphilaand it’s combination with black soldier fly larvae. Life 2023, 13, 234, doi:10.3390/life13010234.
- Feng, J.; Cao, L.; Du, X.; Zhang, Y.; Cong, Y.; He, J.; Zhang, W. Biological Detoxification of Aflatoxin B1 by Enterococcus faeciumHB2-2. Foods 2024, 13, 1887, doi:10.3390/foods13121887.
- Qin, X.; Su, X.; Tu, T.; Zhang, J.; Wang, X.; Wang, Y.; Wang, Y.; Bai, Y.; Yao, B.; Luo, H.; et al. Enzymatic Degradation of Multiple Major Mycotoxins by Dye-Decolorizing Peroxidase from Bacillus subtilis. Toxins 2021, 13, 429, doi:10.3390/toxins13060429.
- Tegtmeier, D.; Hurka, S.; Klüber, P.; Brinkrolf, K.; Heise, P.; Vilcinskas, A. Cottonseed Press Cake as a Potential Diet for Industrially Farmed Black Soldier Fly Larvae Triggers Adaptations of Their Bacterial and Fungal Gut Microbiota. Front. Microbiol. 2021, 12, 634503, doi:10.3389/fmicb.2021.634503.
- Callegari, M.; Jucker, C.; Fusi, M.; Leonardi, M.G.; Daffonchio, D.; Borin, S.; Savoldelli, S.; Crotti, E. Hydrolytic Profile of the Culturable Gut Bacterial Community Associated With Hermetia illucens. Front. Microbiol. 2020, 11, 1965, doi:10.3389/fmicb.2020.01965.
- Nakazato, M.; Morozumi, S.; Saito, K.; Fujinuma, K.; Nishima, T.; Kasai, N. Interconversion of aflatoxin B1 and aflatoxicol by several fungi. Appl. Environ. Microbiol. 1990, 56, 1465-1470, doi:10.1128/aem.56.5.1465-1470.1990.
Please, put in correlation another phrase from the same text (Lines 282-283): “In existing studies, the fungal species in the intestinal tract of BSFL mainly belong to the phylum Ascomycota, including the genera Pichia, Candida, Diutina, Cyberlindnera, Aspergillus, Geotrichum, and Trichosporon [106-110].” Most of these fungi absent in Figure 2. Where are the representatives of Aspergillus genus here?
Response: Thank you for highlighting this question. Figure 2 categorizes Pichia, Candida, Geotrichum, and Trichosporon into a yeast functional group, based on the shared ability of these microorganisms to adsorb AFB1 via their cell wall structures, thereby reducing its bioavailability. Among them, Pichia and Candida are typical yeasts; while Geotrichum and Trichosporon are taxonomically classified as filamentous or dimorphic fungi, they can exhibit yeast-like morphologies in culture and possess toxin adsorption capabilities. Consequently, they are grouped into the same “adsorptive functional unit” in this review. Notably, the original draft merely referenced “yeast” without providing specific taxonomic details, an oversight that has been rectified in lines 540–545 and 593–599 of the revised manuscript. In contrast, Diutina and Cyberlindnera currently lack reports regarding AFB1 detoxification and are thus excluded from this functional group. Meanwhile, Aspergillus species are aerobic microorganisms; even if detected in the gut of BSFL, they cannot sustain their AFB1-degrading activity in the anaerobic intestinal environment. Additionally, Figure 2 only summarizes the microorganisms potentially involved in AFB1 degradation. Thus, the genus Aspergillus is not included in Figure 2.
Table 1: There are some data which are performed in the form of following units “ppm” or “ppb”, whereas the most rest part of information is presented by using traditional units like “μg/mL”, “mg/L” ,“ng/mL” or “μg/kg”. So, I recommend to use single approach to the data presentation. Additionally, he “sp.” in all names of microorganisms should not be italicized.
Response: Thank you for this comment. I have converted all “ppm” and “ppb” in Table 1 into conventional units (ppm to mg/kg; ppb to μg/kg). In addition, the “sp.” component in microbial names has been set in roman type rather than italics. The revised Table 1 is included in the revised manuscript.
There are a lot of places in the text, where the names of authors of articles and numbers of corresponding references are mentioned at the same time: Arun et al. [21], Bovo et al. [20], Gonzalez et al. [23], Petchkongkaew et al. [24], Wang et al. [25], Suo et al. [44], Bosch et al. [96], Purschke et al. [102], Meijer et al. [103], Camenzuli et al. [104]. According to the recommendations of MDPI only numbers of the references should present in the text of the article. Please, remove all names from the text.
Response: Thank you for highlighting the MDPI formatting rule. I have carefully removed all author names from the in-text citations and kept only the reference numbers in brackets (lines 159-163, lines 170-174, lines 176-189, lines 300-301, and lines 367-378 in the revised manuscript). The surrounding sentences have been rephrased where necessary to maintain grammatical correctness. The revised content is as follows:
- For example, three strains, Lactococcus CF_6, Lactobacillussp. CW_3, and Lactobacillus acidophilus CE_4 were isolated from animal excreta, and their AFB1 adsorption rates reached 52.63% to 65.38% [22]. (lines 159-161)
- The AFB1 removal ability of beer fermentation residues and five commercial yeast products was assessed, with AFB1 adsorption rates ranging from 24.0% to 69.4% [21]. (lines 161-163)
- For instance, 11 strains of Bacillus sp. were isolated from pond silt and soil, and the AFB1 degradation rate of these strains ranged from 27.78% to 79.78% after 48 hours [24]. (lines 170-172)
- The Bacillus licheniformisand Bacillus subtilis were isolated from fermented soybeans, which degraded 74% and 85% of AFB1 after 7 days, correspondingly, and inhibited the growth of Aspergillus strains [25]. (lines 172-174)
- For example, BacillusH16v8 and HGD9229 were cocultured, and the degradation rate of the combined strain exhibited 87.7% and 55.3% enhancement, correspondingly, compared with that of H16v8 and HGD9229 alone [26]. (lines 176-189)
- As a case in point, a study experimentally probed the AFB1 degradation ability of BSFL under intestinal sterilized and unsterilized conditions [46]. (lines 300-301)
- For instance, no significant differences in body weight change or mortality of BSFL between a control group and an experimental group fed feeds supplemented with 4600 μg/kg DON, 260 μg/kg OTA, 88 μg/kg AFB1, 17 μg/kg AFB2, 46 μg/kg AFG2 and 860 μg/kg ZEN [112]. (lines 367-370)
- A 97.3% survival rate of BSFL was reported when the larvae were fed a wheat-based diet spiked with 0.5 mg/kg AFB1 [113]. (lines 370-371)
- And in a feed matrix supplemented with 0.415 mg/kg AFB1, BSFL effectively degraded 83%-95.1% of AFB1 while maintaining a high survival rate, with AFB1 concentrations in the larval falling below the threshold of analytical detection (<0.10 µg/kg) [101]. (lines 371-374)
- Another study found no statistically significant disparities in the average survival rate (94%-100%) or average larval fresh weight (172-191 mg/larva) of BSFL between the control group and the groups administered with 8-430 μg/kg AFB1, 170-2000 μg/kg OTA, 280-13000 μg/kg ZEN, 3900-112000 μg/kg DON, or a mixture of mycotoxins [104].(lines 374-378)
References
Totally:Please give the commonly accepted abbreviations instead of complete titles of the Journals in the references.
Response: Thank you for your reminder. All journal names in the reference list have been changed to their standard abbreviations.
Please, check the correctness of the writing of latin names of the living organisms (including the List with all references): the names should be italicized and when you give the genusandspecies of the microorganism the species should start from little letter. Please, see:
Line635: Bacillus
Lines 640: Bacillus licheniformis
Line 643: Bacillus subtilis
Lines 645: Stenotrophomonas Maltophilia; Line 648: Bacillus aryabhatta
Line 658: Bacillus megaterium
Line 664: Aspergillus flavus
Line 669: Pseudomonas
Line 670: Pseudomonas putida
Line 672: Pseudomonas knackmussii
Lines 678, 681: Stenotrophomonas
Line 684: Stenotrophomonas acidaminiphila
Line 688: Rhodococcus erythropolis
Line 691: Rhodococcus
Line 693: Rhodococcus pyridinivorans
Line 695: Rhodococcus turbidus, etc.
Please check all 175 links, almost every one of them has a latin name!
Lines 466-470: there is same situation with latin names of microorganisms. It should be corrected.
Response: Thank you for pointing this out. I have carefully checked and corrected all latin names throughout the manuscript and the 175 references, and all entries have been fixed accordingly.
Reference 9: I recommend to replace it. Bibliography is incomplete in it.
Response: Thank you for highlighting the incompleteness of Ref.9. I have replaced it with the following complete and traceable reference in the revised manuscript.
Reference:
- Wong, J.J.; Hsieh, D.P. Mutagenicity of aflatoxins related to their metabolism and carcinogenic potential. Proc. Natl. Acad. Sci. U.S.A. 1976, 73, 2241-2244, doi:10.1073/pnas.73.7.2241.
The phrase “Supplementary Materials: No more supplement.” can be removed from the article.
Response: Thank you for pointing this out. I have removed the sentence “Supplementary Materials: No more supplement.” from the revised manuscript in line 994.

Round 2
Reviewer 3 Report
Comments and Suggestions for Authors
The authors made a notable work with the article and respectfully responded on all my comments. I am satisfied with the current form of the article.